# Tomosyns attenuate SNARE assembly and synaptic depression by binding to VAMP2-containing template complexes

Marieke Meijer ®[1,5] ✉, Miriam Öttl ®[2,5], Jie Yang ®[3,5] ✉, Aygul Subkhangulova[2], Avinash Kumar[3], Zicheng Feng[3], Torben W. van Voorst ®[2], Alexander J. Groffen[1], Jan R. T. van Weering[1], Yongli Zhang ®[3,4,6] ✉ & Matthijs Verhage ®[1,2,6] ✉

Tomosyns are widely thought to attenuate membrane fusion by competing with synaptobrevin-2/VAMP2 for SNARE-complex assembly. Here, we present evidence against this scenario. In a novel mouse model, tomosyn-1/2 deficiency lowered the fusion barrier and enhanced the probability that synaptic vesicles fuse, resulting in stronger synapses with faster depression and slower recovery. While wild-type tomosyn-1m rescued these phenotypes, substitution of its SNARE motif with that of synaptobrevin-2/VAMP2 did not. Single-molecule force measurements indeed revealed that tomosyn's SNARE motif cannot substitute synaptobrevin-2/VAMP2 to form template complexes with Munc18-1 and syntaxin-1, an essential intermediate for SNARE assembly. Instead, tomosyns extensively bind synaptobrevin-2/VAMP2-containing template complexes and prevent SNAP-25 association. Structure-function analyses indicate that the C-terminal polybasic region contributes to tomosyn's inhibitory function. These results reveal that tomosyns regulate synaptic transmission by cooperating with synaptobrevin-2/VAMP2 to prevent SNAP-25 binding during SNARE assembly, thereby limiting initial synaptic strength and equalizing it during repetitive stimulation.

SNARE-dependent membrane fusion is required for the secretion of most chemical signals[1,2]. As minimal fusion machinery, SNAREs couple their folding and assembly to membrane fusion[3–5]. The fully assembled SNARE complex is a four-helix bundle formed by characteristic SNARE motifs. While SNARE complexes can spontaneously form in vitro with low speed and specificity, their assembly in vivo is controlled by many proteins, such as Munc18s, Munc13s, synaptotagmins, and complexins[2,6–8]. Mutations in the genes encoding SNARE and regulatory proteins lead to a wide range of human diseases, including various neurodevelopmental, cardiovascular and hematological disorders[8–12]. However, many aspects of regulated SNARE assembly remain poorly understood.

Synaptic vesicle (SV) fusion is preceded by a dynamic multi-step process involving docking the SV onto the presynaptic membrane and priming the SV for rapid fusion upon stimulation[13–16]. These steps are associated with distinct assembly intermediates of three synaptic SNAREs: syntaxin-1, SNAP-25, and synaptobrevin-2(syb2)/VAMP2. The template complex constitutes a key intermediate of SNARE assembly[17–23]. In this complex, syntaxin-1 and syb2/VAMP2 bind to the surface of Munc18-1 such that the N-terminal regions of their SNARE motifs are aligned, while their C-terminal regions are kept separated. SNAP-25 specifically and rapidly associates with the template complex, leading to full SNARE assembly[19,24]. It remains unclear how the

[1]Department of Human Genetics, Center for Neurogenomics and Cognitive Research, Amsterdam University Medical Center, 1081HV Amsterdam, The Netherlands. [2]Department of Functional Genomics, Center for Neurogenomics and Cognitive Research, Vrije Universiteit Amsterdam, 1081HV Amsterdam, The Netherlands. [3]Department of Cell Biology, Yale School of Medicine, New Haven, CT 06511, USA. [4]Department of Molecular Biophysics and Biochemistry, Yale University, New Haven, CT 06511, USA. [5]These authors contributed equally: Marieke Meijer, Miriam Öttl, Jie Yang. [6]These authors jointly supervised this work: Yongli Zhang, Matthijs Verhage. ✉e-mail: m.meijer@vu.nl; jie.yang.jy546@yale.edu; Yongli.zhang@yale.edu; m.verhage@vu.nl

template complex is regulated by other proteins and affects the strength and short-term plasticity properties of synapses (e.g., facilitating/'tonic' or depressing/'phasic') which help determine the computational properties of brain circuits[25,26].

Tomosyns play important roles in vesicle fusion[27], but their mechanism of action remains unclear. Tomosyn isoforms share a conserved structure containing a large N-terminal WD40 repeat domain and a small C-terminal SNARE motif[28,29]. In cell-free assays, the tomosyn SNARE motif competes with the homologous SNARE motif of syb2/VAMP2, resulting in a tomosyn SNARE complex that is expected to be non-fusogenic as tomosyns lack a transmembrane domain present in VAMP2[30–33]. Indeed, tomosyns inhibit exocytosis in many secretory cell types[30,34–37], including neurons at neuromuscular junctions in nematode and fly[38–41] and mammalian central and peripheral neurons[42–44], with a few exceptions hinting at an additional positive role[45–47]. Mutations in tomosyns are associated with neurodevelopmental disorders and thrombosis[11,48–54], suggesting an important and conserved function in different secretory tissues. However, several observations are inconsistent with the concept of tomosyns as competitive inhibitors of syb2/VAMP2. In mouse platelets and yeast, tomosyns (-orthologs) exert positive roles in secretion[37,55,56]. In addition, the N-terminal WD40 domain was found to be required for its inhibitory function[41,57,58]. Furthermore, NSF/αSNAP rapidly disassemble the syntaxin-1/SNAP-25 binary complex required for the formation of the tomosyn SNARE complex[31], arguing against its proposed role in SNARE assembly in vivo. These observations challenge the concept of tomosyns as competitive inhibitors of SNARE complex formation.

This study addresses these issues by combining secretion assays in living neurons, measurements of the energy barrier for fusion, and single-molecule force measurements using optical tweezers. Loss of tomosyn expression lowers the energy barrier for fusion which results in initially stronger but fast-depressing synapses. Expression of hybrid constructs where syb2/VAMP2 replaced corresponding domains in tomosyn fails to restore these synaptic defects. In line with these observations, single-molecule force measurements show that the tomosyn SNARE motif fails to form a template complex with Munc18-1 and syntaxin-1, indicating that tomosyns do not compete with syb2/VAMP2 during Munc18-1-chaperoned SNARE assembly. Instead, tomosyns bind to syb2/VAMP2-containing template complexes and block integration of SNAP-25, thereby lowering the fusion rate of synaptic vesicles, and limiting synaptic depression.

## Results

### A conditional KO mouse for both tomosyn paralogs

Due to prior complications with sub-lethal phenotypes and bleeding risk in constitutive mutant mouse models[44,56], we generated a conditional double floxed mouse model in which both tomosyn paralogs can be inactivated by Cre-mediated recombination at lox sites flanking exon 2 of tomosyn-1 (*Stxbp5*) and exon 3 of tomosyn-2 (*Stxbp5l*) (Supplementary Fig. 1a). Hippocampal neurons cultured from newborn mutant mice were infected with lentiviruses expressing EGFP-tagged Cre to induce recombination (cDKO), while inactive Cre (lacking the catalytic domain) was used as control[59]. Immunoblotting confirmed that floxed neurons infected with Cre lacked a band at the expected position of 130 kDa for tomosyn-1 (Supplementary Fig. 1b). The conditional inactivation of tomosyn-2 has been validated before[47]. Depleting neurons of both tomosyns did not affect the levels of the neuronal SNARE proteins syntaxin-1 and syb2/VAMP2 but resulted in a 20% reduction of SNAP-25 levels (Supplementary Fig. 1b, c).

To investigate the role of tomosyns in synapse function, single hippocampal neurons were cultured on astrocyte microdots, driving neurons to form abundant synapses onto themselves, resulting in a one-neuron circuit[60,61]. Immunostainings showed a punctate pattern of endogenous tomosyn-1, similar to synaptophysin-1 puncta, confirming

the synaptic enrichment of tomosyns[40,41,45,62,63] (Supplementary Fig. 1d). We quantified the tomosyn staining intensity within the synaptophysin-1 puncta, which confirmed the effectiveness of Cre in preventing expression of tomosyns in cDKO neurons (Supplementary Fig. 1d, e). Knockdown of tomosyn-1 in cultured mouse neurons has recently been reported to reduce dendritic arborization[64]. We quantified neuronal morphology in our single-cell assay using the automated image analysis routine SynD[65]. Two-week old autaptic cDKO neurons displayed normal dendritic length and complexity as well as synapse density (Supplementary Fig. 1f–h). This was also the case at an earlier time point during development (Supplementary Fig. 1i–l). Hence, autaptic hippocampal neurons showed no evidence of altered dendritic branching and synapse formation in the absence of both tomosyns.

### Tomosyn cDKO synapses are stronger and depress faster

Next, we performed whole-cell patch-clamp recordings on autaptic hippocampal neurons to assess the role of tomosyns in synaptic transmission. cDKO neurons displayed a 5-fold increase in the frequency of spontaneous vesicle release events (miniature excitatory postsynaptic currents, mEPSCs), while the amplitude of these events was normal (Fig. 1a–c). The amplitudes of evoked EPSCs were larger in cDKO neurons (Fig. 1d, e), while their kinetics were normal (Supplementary Fig. 2a–c). We performed paired-pulse recordings and calculated the paired-pulse ratio (PPR), which inversely correlates with the initial synaptic release probability (Pr)[66,67]. cDKO neurons showed reduced PPRs at all intervals tested, indicative of an increased initial release probability in the absence of tomosyns (Fig. 1f, g and Supplementary Fig. 2d). Moreover, cDKO neurons displayed pronounced depression during short trains of action potentials, while control neurons largely maintained their synaptic strength (Fig. 1h–k and Supplementary Fig. 2e, f). Thus, tomosyns lower the release probability of synapses, thereby reducing their initial strength and limiting synaptic depression.

Previous studies have demonstrated that synaptic perturbations have different impact on single neurons versus neuronal networks[68–70]. Therefore, we analyzed synaptic transmission in cDKO versus control synapses also in micro-networks of 3-10 neurons. Consistent with our findings in single neurons, cDKO neurons in micro-networks displayed a strong increase in mEPSC frequency and normal mEPSC amplitude (Supplementary Fig. 3a–c) and more pronounced synaptic depression during short trains of action potentials applied by local field stimulation (Supplementary Fig. 3d–g). Hence, loss of tomosyns yielded identical results in single neurons and micro-networks.

In chromaffin cells, tomosyn-1 overexpression shifts the calcium-dependence of exocytosis[35]. To test the calcium-dependence of synaptic responses, we applied paired pulses to autaptic neurons across a range of external calcium concentrations ($[Ca^{2+}]_e$) (Supplementary Fig. 4a). The initial EPSC size and PPR depended on $[Ca^{2+}]_e$, as expected. At each concentration, PPRs were lower in cDKO neurons compared to control neurons (Supplementary Fig. 4b). EPSC amplitudes were normalized to flanking responses in standard 2 mM $[Ca^{2+}]_e$ and fitted with a Hill function (Supplementary Fig. 4c). The apparent calcium affinity was found to be higher (i.e., decreased $K_d$ for calcium) in cDKO neurons ($K_d = 1.3 \pm 0.1$ mM) compared to control neurons ($K_d = 1.7 \pm 0.1$ mM) (Supplementary Fig. 4d, left). The Hill coefficient, a measure of calcium cooperativity, was unchanged (Supplementary Fig. 4d, right). These results suggest that tomosyns reduce the $Ca^{2+}$-sensitivity of synaptic responses.

### Tomosyns increase the energy barrier for vesicle fusion

Application of hypertonic solutions is used to release SVs from the readily releasable pool (RRP) independent of $Ca^{2+}$ and action potentials[71]. The total RRP, as defined by the response to 500 mM sucrose, was similar in tomosyn cDKO neurons and control neurons (Fig. 2a, b). However, the fraction of the RRP released by a single EPSC,

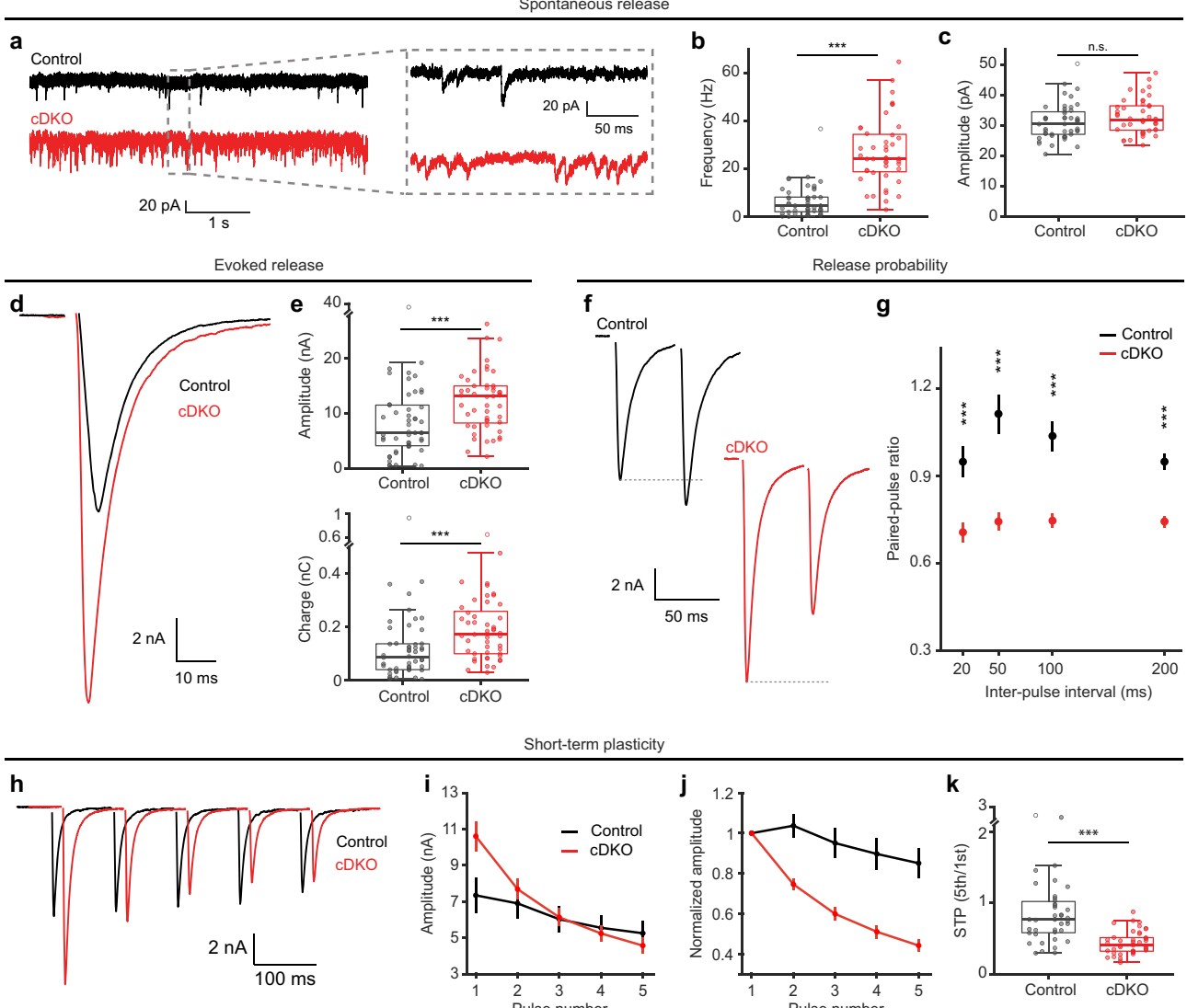

**Fig. 1 | Tomosyn cDKO synapses are stronger and depress faster. a–c** Analysis of spontaneous vesicle release in autaptic hippocampal neurons. Control n = 47/6, cDKO n = 44/6. **a** Example traces of miniature EPSCs (mEPSCs). **b** The frequency of mEPSCs (***$p < 0.0001$). **c** mEPSC amplitude ($p = 0.0832$). **d–e** Analysis of evoked synaptic transmission. Control n = 50/6, cDKO n = 49/6. **d** Example traces of evoked synaptic transmission. **e** Quantification of evoked EPSC amplitude (***$p = 0.0001$) and charge (***$p = 0.0002$). See Supplementary Fig. 2a–c for quantifications of EPSC kinetics. **f, g** Release probability was tested by applying paired pulses with multiple inter-pulse intervals (IPIs). 20 ms IPI: control n = 45/6, cDKO n = 47/6. 50 ms IPI: control n = 37/6, cDKO n = 42/6. 100 ms IPI: control n = 39/6, cDKO n = 44/6. 200 ms IPI: control n = 47/6, cDKO n = 47/6. **f** Example traces of paired pulses stimulated with a 50 ms IPI. **g** The paired-pulse ratio (PPR) was calculated by dividing the amplitude of the second pulse by the amplitude of the first pulse; (***$p < 0.0001$). See Supplementary Fig. 2d for boxplots per IPI. **h–k** Short-term plasticity (STP) was analyzed by stimulation of five consecutive pulses at 10 Hz. Control n = 47/6, cDKO n = 47/6. **h** Example traces. **i** Absolute EPSC amplitudes over the 5 pulses. **j** STP illustrated by normalizing amplitudes to the first pulse. **k** STP quantified by the ratio of the fifth pulse over the first pulse; (***$p < 0.0001$). N = cells/independent cultures. In (**b, c, e, k**), boxplots display median (center), upper and lower quartiles (box bounds) and whiskers to the last datapoint within 1.5x interquartile range. In (**g, i, j**), data are presented as mean ± SEM. A one-way ANOVA tested the significance of adding experimental group as a predictor, see Supplementary Table 1. Abbreviations: n.s. (not significant). See also Supplementary Figs. 1–4. Source data are provided as a Source Data file.

a measure of the vesicular release probability ($P_{ves}$), was increased in cDKO neurons (Fig. 2c). $P_{ves}$ is regulated by the presynaptic $Ca^{2+}$-influx and the $Ca^{2+}$-sensitivity of the membrane fusion reaction, two highly regulated properties that have a large impact on synaptic plasticity. In addition, $Ca^{2+}$-independent aspects of membrane fusion contribute to $P_{ves}$ such as the lipid composition of both membranes and the organization of the fusion machinery prior to the arrival of action potentials, that together determine how likely membranes fuse, also referred to as 'fusogenicity'[72,73]. Application of submaximal sucrose concentrations can be used to probe this fusogenicity and, in contrast to $P_{ves}$ measurements, excludes the modulation of $Ca^{2+}$-dependent aspects[74,75]. In the absence of tomosyns, 250 mM sucrose released a much larger fraction of the RRP (Fig. 2d). This indicates that tomosyns reduce the fusogenicity of SVs without affecting the size of the primed SV pool.

Fusogenicity is inversely related to the amount of energy required to merge vesicle- and plasma membrane during stimulation[2,76–78]. Molecular changes in the secretion machinery are known to alter this barrier prior to stimulation, so that the (remaining) energy required during stimulation is also altered and vesicles are more or less likely to fuse (fusogenic). The energy barrier during stimulation can be derived from the fusion rate (which is measured directly) using the Arrhenius equation[75,79]. We have previously shown that fusion rates can be obtained from fitting hypertonic sucrose responses to a minimal

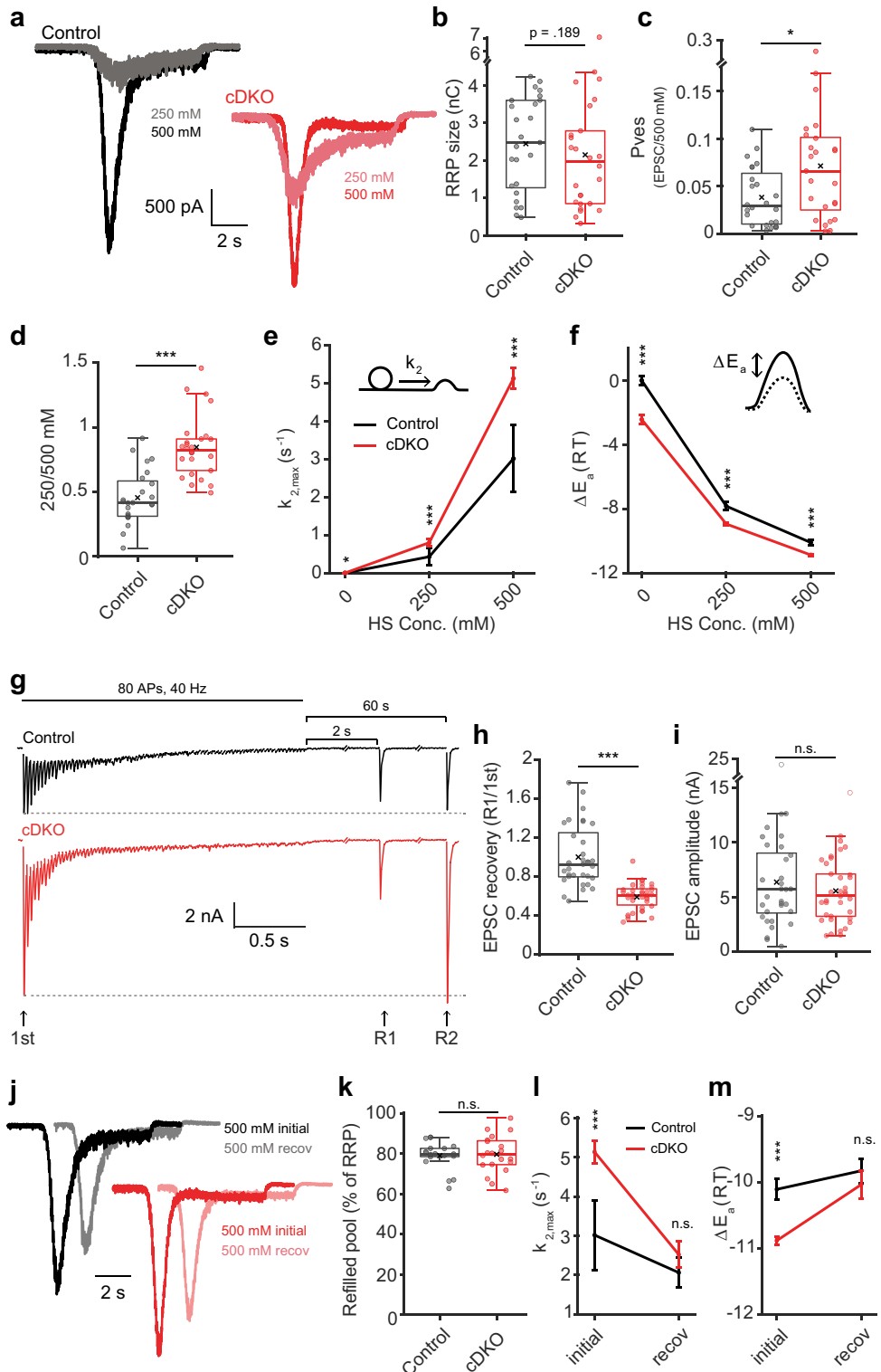

**Nature Communications** | (2024)15:2652

vesicle state model (illustrated in Supplementary Fig. 5a) and can be used to calculate the fusion energy barrier[75,80]. We applied this analysis to the sucrose responses acquired above (Fig. 2). Neurons lacking tomosyns have higher maximal fusion rates compared to controls at 250 and 500 mM sucrose (Fig. 2e), corresponding to a respective shift in the fusion barrier of ~−1.11RT and ~−0.77 RT compared to controls (Fig. 2f). Data fitting also yielded estimates for the RRP size, which confirmed the unchanged RRP size and increased $P_{ves}$ (Supplementary Fig. 5b–d), as well as the priming and de-priming rates of SVs. These rates were not significantly altered in the absence of tomosyns (but

power was rather low due to large data variation, Supplementary Fig. 5e, f). Overall, these results show that tomosyns do not influence the number of releasable vesicles, but substantially decrease the likeliness that vesicles fuse during stimulation (fusogenicity), by increasing the energy barrier for fusion.

**Tomosyns promote rapid recovery after intense stimulation**
To test whether tomosyns impact the recovery of synaptic strength after intense activity, single recovery pulses were given 2 s (R1) and 60 s (R2) after exhaustive high-frequency stimulation (Fig. 2g).

**Fig. 2 | Tomosyns increase the energy barrier for vesicle fusion. a** Example responses to sucrose application. **b** RRP released by 500 mM sucrose. Control n = 25/6, cDKO n = 26/6; p = 0.1894. **c** The fraction of the RRP released by an EPSC. Control n = 24/6, cDKO n = 25/6; *p = 0.0137. **d** The fraction of the RRP released by 250 mM sucrose. Control n = 25/6, cDKO n = 26/6; ***p < 0.0001. **e, f** Sucrose traces were fitted to a minimal vesicle state model. 0 mM (rest): control n = 13/6, cDKO n = 7/6; 250 mM: control n = 12/6, cDKO n = 17/6; 500 mM: control n = 16/6, cDKO n = 18/6. See Supplementary Fig. 5a. **e** Fitted maximal fusion rate constant ($k_{2,max}$(s$^{-1}$)); *p = 0.02436 (0 mM); ***p = 0.0001 (250 mM); ***p < 0.0001 (500 mM). **f** The change in activation energy (ΔEa) in relation to control neurons at rest, calculated from $k_{2,max}$(s$^{-1}$), reflecting the change in the energy barrier upon stimulation. ***p = 0.0004 (0 mM); ***p = 0.0001 (250 mM); ***p < 0.0001 (500 mM). **g–i** RRP-depleting stimulus followed by recovery pulses after 2 (R1) and 60 (R2) seconds.

Control n = 33/6, cDKO n = 38/6. **g** Example traces. **h** Fraction of 1$^{st}$ EPSC amplitude recovered at R1; ***p < 0.0001. **i** Absolute R1 amplitude; p = 0.4889. See Supplementary Fig. 5j-k for R2. **j–m** A second 500 mM sucrose applications (recov) was given after 30 s. **j** Normalized example traces. **k** Refilled pool as percentage of initial RRP. Control n = 19/6, cDKO n = 20/6, p = 0.6347. **l** The fusion rate and (**m**) the change in the activation energy. Initial: corresponds to (**e–f**). Recov: control n = 12/6, cDKO n = 11/6; p = 0.4030 ($k_{2,max}$(s$^{-1}$)); p = 0.4182 (ΔEa). N = cells/independent cultures. In (**b, c, d, h, i, k**), boxplots display median (center), upper and lower quartiles (box bounds) and whiskers to the last datapoint within 1.5x interquartile range. In (**e, f, l, m**), data report mean ± SEM. A one-way ANOVA tested the significance of adding experimental group as a predictor, see Supplementary Table 1. Abbreviations: n.s. (not significant), HS (hypertonic sucrose). See also Supplementary Figs. 5, 6. Source data are provided as a Source Data file.

This also allowed for an alternative measure of the RRP by back-extrapolating the cumulative charge released by the stimulus train to the y-intercept (Supplementary Fig. 5g)[81]. Again, the RRP size was similar in control and cDKO neurons (Supplementary Fig. 5h). In addition, the refilling rate of SVs was unchanged, measured by the slope of the back-extrapolation line (Supplementary Fig. 5i). Nevertheless, while control neurons recovered to ~100% of their initial EPSC amplitude within 2 s, cDKO neurons only reached ~60% (Fig. 2h). Notably, the absolute EPSC amplitude was similar in control and cDKO neurons at this point (Fig. 2i). During the second recovery pulse given after a full minute, cDKO neurons had restored their initially larger EPSC size (Supplementary Fig. 5j, k). To further explore this recovery phenotype, we performed a dual sucrose experiment to measure the extent of calcium-independent RRP recovery (Fig. 2j). Both control and cDKO neurons recovered their RRP to 80% within 30 s (Fig. 2k). The fusion rate and energy barrier after 30 s recovery were similar to control levels, in stark contrast to the initial values (Fig. 2l, m). Together, these results show that while SV refilling is normal, cDKO neurons take longer to recover their enhanced synaptic strength, indicating that tomosyns promote rapid recovery after intense stimulation.

### Tomosyns have no major impact on the docked SV pool

Nematode and fly tomosyn mutants were shown to have more docked vesicles, suggesting that invertebrate tomosyn inhibits SV docking[39,41]. To test whether tomosyns affect SV docking in mammalian neurons, we performed high-pressure freeze electron microscopy on low density neuronal cultures grown on glia-covered sapphires. cDKO synapses showed a trend towards a reduced number of SVs contacting the active zone compared to controls (Supplementary Fig. 6a, b). However, when Tomosyn was reintroduced in cDKO synapses by lentiviral expression (+WT group), the number of docked vesicles was significantly reduced compared to controls but not cDKO. In a few synapse cross sections (control: 26 of 130 synapses, cDKO: 11 of 88 synapses, cDKO +WT: 12 of 126 synapses), vesicles contacting the plasma membrane outside the active zone were observed but their number was not different between groups (Supplementary Table 1). The total number of vesicles per synaptic profile, active zone length and vesicle distribution was normal (Supplementary Fig. 6c–e). These data show that the increased synaptic strength in tomosyn cDKO neurons is unlikely to be explained by an increased docked/primed vesicle pool in mammalian neurons.

### A syb2/VAMP2-hybrid fails to mimic tomosyn's inhibitory function

The C-terminal SNARE motif of tomosyns is thought to compete with the corresponding motif in syb2/VAMP2 during neuronal SNARE assembly (Fig. 3a). In such a scenario, tomosyns would be equally functional with the syb2/VAMP2 SNARE motif. To test this, a hybrid tomosyn was cloned in which the C-terminal SNARE motif of tomosyn-1m, the most abundant tomosyn isoform in the hippocampus[82], was replaced with the SNARE motif of syb2/VAMP2 (Fig. 3b). Expressing

wild-type tomosyn in cDKO neurons restored normal synaptic tomosyn levels, while hybrid tomosyn was expressed at an even higher level (Fig. 3c and Supplementary Fig. 7). Neuronal morphology was not altered by the expression of either variant and both variants targeted normally to synapses (Supplementary Fig. 7d). Expression of wild-type tomosyn-1m restored the enhanced synaptic depression during a short 10 Hz train in cDKO neurons (Fig. 3d–f) as well as the defect in synaptic recovery after high-frequency stimulation (Fig. 3g, h). Hybrid tomosyn rescued these synaptic changes only to a limited extent (Fig. 3d–h), despite the high expression levels of this variant. All groups restored their initial EPSC size within 90 s after 10Hz-induced depression (Supplementary Fig. 7f). These observations are inconsistent with the simple competition model, in which tomosyns regulate synaptic transmission by competing with syb2/VAMP2 in SNARE assembly.

### The tomosyn SNARE motif forms a stable ternary complex

The competition model was based on the observation that the SNARE domain of tomosyns, like syb2/VAMP2, forms a stable ternary complex with the SNARE domains of syntaxin-1 and SNAP-25 when the isolated domains are mixed in solution[32]. However, SNARE assembly in vivo is chaperoned by Munc18-1 and other regulatory proteins, which may alter the SNARE assembly pathway observed in vitro. Consequently, it is unknown whether such tomosyn SNARE complexes are assembled under physiological conditions. We previously developed a single-molecule assay based on high-resolution optical tweezers to characterize SNARE assembly[8,19,83–85]. Here, we adopted this assay to characterize the energetics and kinetics of tomosyn SNARE assembly by pulling on a single pre-assembled tomosyn SNARE complex. To this end, we connected the C-termini of syntaxin and tomosyn to two beads (Fig. 4a). The beads were held in two optical traps as force and displacement sensors. When the tomosyn SNARE complex was being pulled by moving one trap relative to the other, the extension and tension of the protein-DNA tether were measured to report conformational changes of the complex and their associated energies. We first tested whether in this assay, tomosyn SNARE complexes are formed, as previously shown in solution, and quantitatively compared their energetics and kinetics to those of syb2/VAMP2 SNARE complexes. Subsequently we tested competition between tomosyn and syb2/VAMP2 in Munc18-1-chaperoned SNARE assembly.

In the first set of experiments, tomosyn and syntaxin were crosslinked at their -6 layers through a disulfide bond (Fig. 4a, X-6) in a single pre-assembled tomosyn SNARE complex containing SNAP-25. The crosslinking enable the SNARE complex to undergo multiple rounds of disassembly/assembly and stepwise folding and unfolding transitions under equilibrium conditions[86]. We began by pulling the pre-assembled complex in the absence of Munc18-1 (Fig. 4c, state 1'). The tomosyn SNARE complex unfolded stepwise with characteristic intermediate states and kinetics as seen in the force-extension curve (FEC, Fig. 4b, FEC#1, gray curve). This FEC is equivalent to a phase diagram revealing the states and their transitions of the tomosyn SNARE complex as a function of force and extension[84]. The FEC is

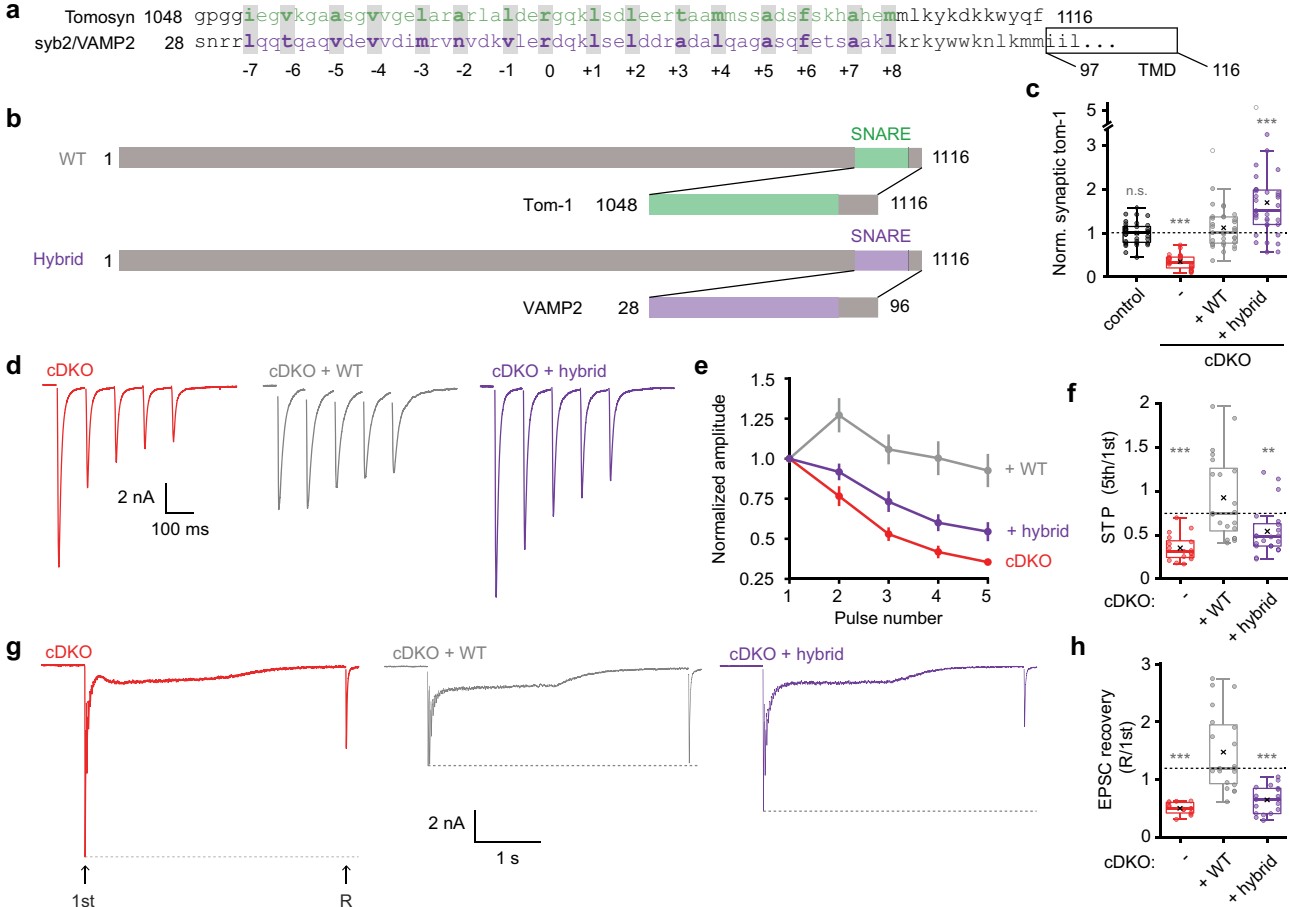

**Fig. 3 | A tomosyn-VAMP2 hybrid fails to fully restore tomosyns' inhibitory function. a** Comparison of the amino acid sequences of the syb2/VAMP2 and tomosyn R-SNARE motifs. TMD = transmembrane domain. **b** Schematic representation of WT tomosyn-1m and the hybrid mutant, in which we replaced the C-terminus of tomosyn with the corresponding region in VAMP2. Numbers correspond to amino acids. **c** Synaptic tomosyn-1 expression from immunostainings, normalized to control within each neuronal culture. Control n = 35/3, cDKO n = 30/3, +WT n = 33/3, +Hybrid n = 35/3; p = 0.3902 (control vs +WT); ***p < 0.0001 (cDKO vs +WT), ***p = 0.0002 (+hybrid vs +WT). **d**–**f** Analysis of short-term plasticity (STP) by stimulating neurons with five pulses at 10 Hz. cDKO n = 17/4, +WT n = 21/4, +Hybrid n = 21/4. **d** Example traces. **e** Amplitudes were normalized to the first pulse of the train. **f** STP quantified by the ratio of the fifth pulse over the first pulse; ***p < 0.0001 (cDKO vs +WT), **p = 0.0023 (+hybrid vs +WT). **g**, **h** Recovery of the

first EPSC amplitude (1st) was tested by high-frequency stimulation (80 pulses at 40 Hz) followed by a recovery pulse after 2 s (R). cDKO n = 14/4, + WT n = 20/4, +Hybrid n = 18/4. **g** Example traces. **h** The amplitude of the recovery pulse was divided by the first amplitude of the train; ***p < 0.0001 (cDKO vs +WT), ***p < 0.0001 (+hybrid vs +WT). N = cells/independent cultures. In (**c**, **f**, **h**), boxplots display median (center), upper and lower quartiles (box bounds), and whiskers to the last datapoint within 1.5x interquartile range. In (**e**), data are presented as mean ± SEM. A one-way ANOVA tested the significance of adding experimental group as a predictor, see Supplementary Table 1. For post-hoc comparison to + WT group, p value thresholds (*<0.05; **<0.01;***<0.001) were adjusted with a Bonferroni correction (α/number of tests). Abbreviations: TMD (transmembrane domain), n.s. (not significant). See also Supplementary Fig. 7. Source data are provided as a Source Data file.

quantitatively similar to that of the syb2/VAMP2 SNARE complex[86], which allowed us to identify the intermediate states. These are schematically depicted in Fig. 4c and represent unfolding of the C-terminal half of the tomosyn SNARE motif into a half-zippered SNARE bundle (state 2′), unfolding and dissociation of the N-terminal half of the tomosyn SNARE motif from a binary t-SNARE complex of syntaxin-1:SNAP-25 (state 3′)[87], and complete unfolding of the syntaxin SNARE motif and subsequent dissociation of SNAP-25 (state 4′). Relaxing the unfolded tomosyn-Syx conjugate resulted in a featureless FEC (Fig. 4b, FEC#1, black curve), which indicates that the remaining SNAREs did not refold after SNAP-25 dissociated. Thus, the tomosyn SNARE motif barely associates with syntaxin-1 alone, but instead requires a preassembled syntaxin/SNAP-25 dimer[8,87].

We then held the tomosyn SNARE complex at constant trap separations or mean forces and detected its folding and unfolding transitions with high spatiotemporal resolution (Fig. 4d). These trajectories revealed reversible transitions among the folded four-helix bundle (state 1′), the half-zippered SNARE bundle (state 2′), and the

unzipped tomosyn SNARE motif (state 3′). Hidden-Markov analyses of the trajectories demonstrated that the transitions are sequential and direct transitions between states 1′ and 3′ are negligible[88,89]. Thus, the folding and unfolding transitions of the tomosyn SNARE complex can be divided into the transitions of the C-terminal domain (CTD) and the N-terminal domain (NTD) (Fig. 4a). The CTD transition is frequent, indicating a great transition rate, while the NTD transition is ~400-fold slower (Fig. 4d). Based on the measured force ranges for both transitions, we estimated unfolding energies of 23 (±1, SEM) $k_BT$ and 35 (±2) $k_BT$ for the CTD and the NTD, respectively, compared with the corresponding CTD energy of 22 $k_BT$ and NTD energy of 38 $k_BT$ for the syb2/VAMP2 SNARE complex[86]. Overall, in the absence of chaperones such as Munc18-1, the tomosyn SNARE complex folds and unfolds similarly to the syb2/VAMP2 SNARE complex (Fig. 4d, bottom trace), including their similar energetics[83,86]. However, the NTD of the tomosyn SNARE complex folds much slower than that of the syb2/VAMP2 SNARE complex. These results imply that the tomosyn SNARE motif is energetically competent, but kinetically

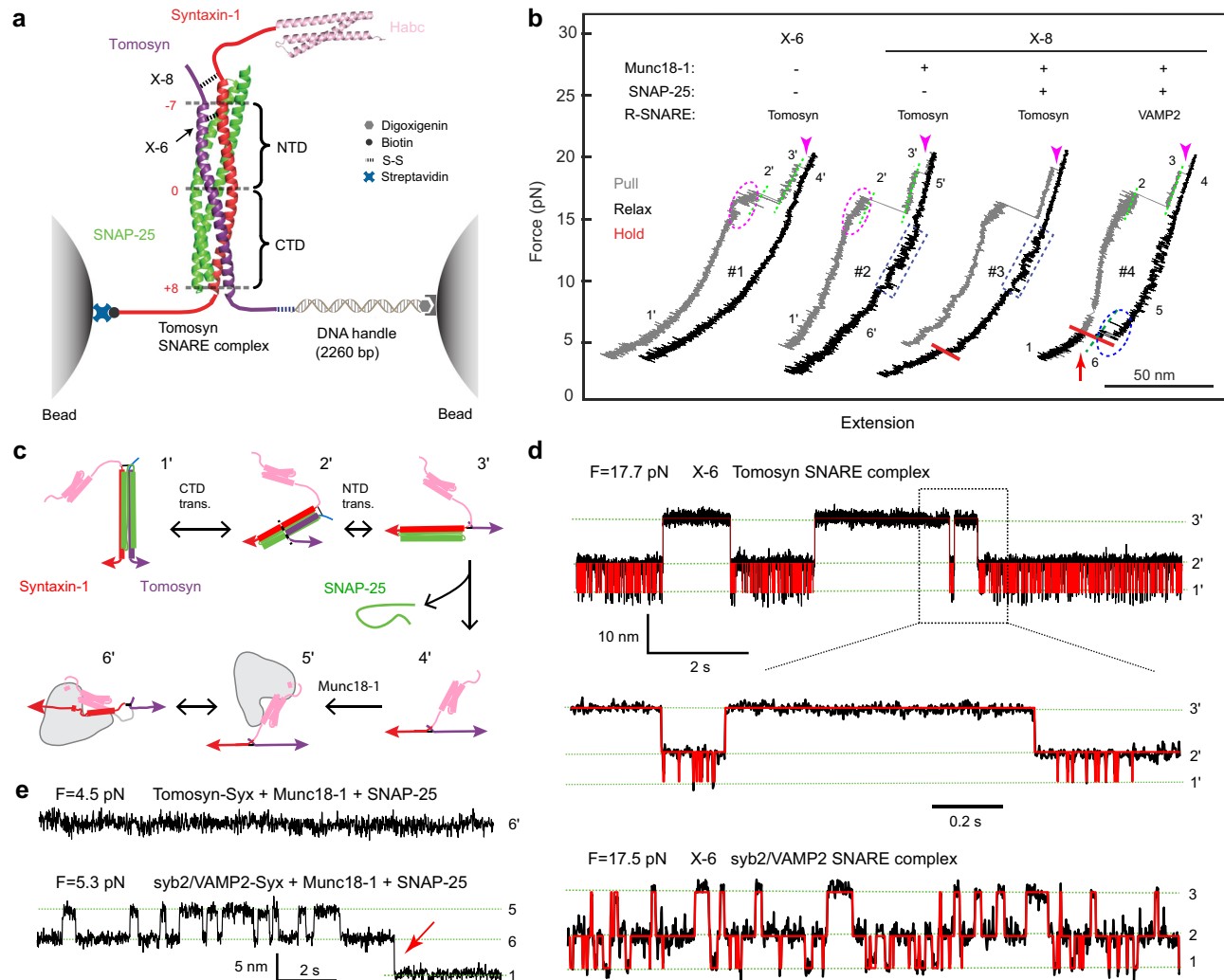

**Fig. 4 | The SNARE motif of tomosyns forms a stable SNARE complex with syntaxin-1 and SNAP-25, but fails to form a template complex with Munc18-1 and syntaxin-1. a** Experimental setup to pull a single tomosyn SNARE complex (PDB ID: 1URQ) using optical tweezers. Tomosyn and syntaxin-1 (Syx) molecules in the pre-assembled complex were attached to two polystyrene beads via a DNA handle at their C-termini and crosslinked at either the -6 layer (X-6) or the −8 layer (X-8). The central ion layer divides the parallel helix bundle into the N-terminal domain (NTD) and the C-terminal domain (CTD). Munc18-1 and SNAP-25 may be added to the solution either alone or together to test template complex formation or SNARE assembly. The presence or absence of these proteins free in solution is indicated by "+" or "−", respectively. **b** Force-extension curves (FECs) obtained by pulling (gray) or relaxing (black) the tomosyn-Syx conjugate at a speed of 10 nm/s or holding it at a constant mean force or trap separation (red). Different FEC regions, indicated by green dashed lines, are labeled by the numbers of their associated states in panel **c**. The dashed magenta oval, blue rectangles and blue oval mark reversible folding and unfolding transitions of tomosyn SNARE four-helix

bundle (see Fig. 4d), Munc18-1-bound open syntaxin and the template complex, respectively. Red arrows indicate events of SNAP-25 binding to the template complex. Magenta arrow heads mark the unfolding of t-SNARE complexes and the accompanying dissociation of SNAP-25 from the pre-assembled SNARE complexes. **c** Schematic diagrams of different states involved in SNARE folding/assembly and Munc18-1 association, including the fully assembled tomosyn SNARE complex (1′), the half-zippered four-helix bundle (2′), the unzipped tomosyn (3′), the unfolded SNARE motifs (4′), the unfolded SNARE motifs with Munc18-1 bound to the N-terminal region of syntaxin-1 (5′), and the Munc18-1-bound open syntaxin (6′). See Fig. 5c for similar states with tomosyn replaced by VAMP2. **d** Extension-time trajectories at the indicated constant forces (F) showing three-state folding and unfolding transitions of the tomosyn or syb2/VAMP2 SNARE complex. Red curves represent the idealized transitions derived from hidden Markov modeling. Green dashed lines mark the average extensions of the associated states labeled on the right. **e** Extension-time trajectories of different SNARE conjugates at constant low forces.

incompetent for competing with syb2/VAMP2 to form the SNARE complex.

## Tomosyn does not compete with syb2/VAMP2 in the template complex

We and others have shown that the physiological SNARE assembly likely follows a template complex-dependent Munc18-1-chaperoned pathway (depicted in Fig. 5a, c, state 7), which is significantly different from the above t-SNARE complex dependent pathway[8,17,19,23]. To examine whether tomosyn can replace syb2/VAMP2 in the template complex, we crosslinked tomosyn and syntaxin-1 at their -8 layers (X-8)

as in our previous assay[19]. Crosslinking at this alternative site opens the closed syntaxin-1 conformation bound by Munc18-1 but minimally perturbs the template complex. As before, we pulled the pre-assembled tomosyn SNARE complex to stepwise unfold the tomosyn-Syx conjugate, leading to dissociation of the SNAP-25 molecule (Fig. 4b, FEC#2, gray curve, Fig. 4c, states 1′-4′). The conjugate was then slowly relaxed in the presence of 2 μM Munc18-1 in solution to examine template complex formation. Munc18-1 first bound to form open syntaxin-1 in a reversible manner in the force range of 10-17 pN as previously observed (Fig. 4b, FEC#2 in blue dashed rectangle; Fig. 4c, state 6′)[19]. However, no further folding transition was detected at a

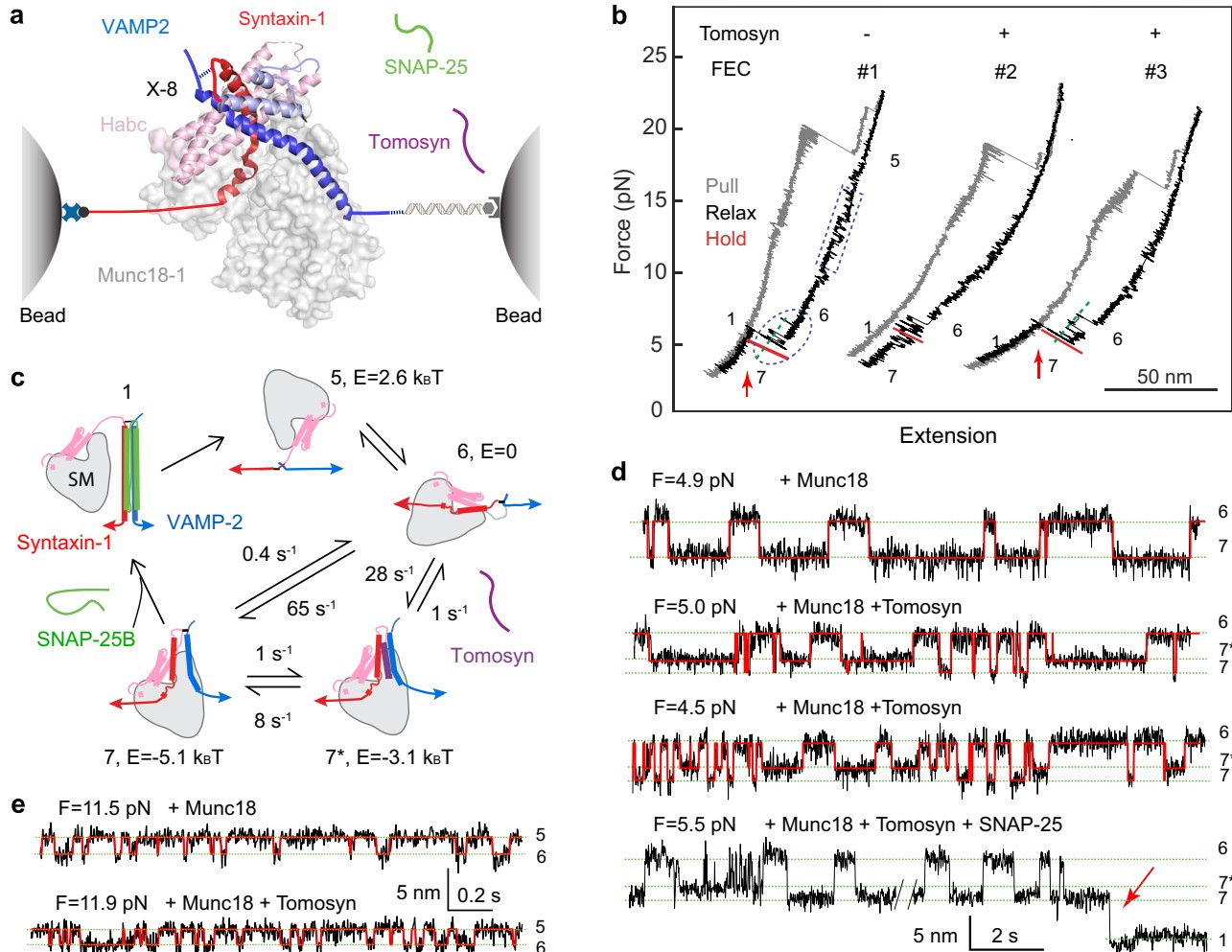

**Fig. 5 | The tomosyn SNARE motif binds to the template complex to block its association with SNAP-25. a** Experimental setup to investigate the effect of the tomosyn SNARE motif on the folding of the synaptic template complex Munc18-1:syntaxin-1:VAMP2 (PDB ID 1SFC). **b** FECs of the VAMP2-Syx conjugate in the presence of 2 μM Munc18-1, 60 nM SNAP-25, and 0 (#1) or 2 μM (#2 and #3) tomosyn SNARE motif in the solution. The presence or absence of these proteins free in solution is indicated by "+" or "−", respectively. The dashed rectangle and oval mark reversible folding and unfolding transitions of Munc18-1-bound open syntaxin and the template complex, respectively. Red arrows indicate events of SNAP-25 binding to the template complex. Green dashed lines indicate the template complex state (state 7). **c** Schematic diagrams of different states and their associated energies (E) and transition rates. These states include the fully assembled SNARE complex with

or without Munc18-1 bound to the Habc domain of syntaxin (state 1), unfolded SNARE motifs (5), open syntaxin (6), the template complex (7), and the tomosyn-bound template complex (7*). In all these states, Munc18-1 can bind to the N-terminal region of syntaxin-1, including its Habc domain. **d** Extension-time trajectories at the indicated constant forces (F) showing folding and unfolding transitions between the states labeled on the right. The red curves represent the idealized transitions derived from hidden Markov modeling. The red arrow designates the SNAP-25 binding event. **e** Extension-time trajectories at constant forces showing folding and unfolding transitions of the Munc18-1-bound open syntaxin-1 in the absence (top) or presence of the tomosyn SNARE motif in the solution. See also Supplementary Fig. 9.

lower force range (Fig. 4b, FEC#2, black FEC), even when the conjugate was held at a low force for a long time (Fig. 4b, FEC#3, red region), suggesting that the tomosyn SNARE motif cannot substitute the syb2/VAMP2 SNARE motif to form the template complex. In the absence of the template complex as a key intermediate for Munc18-1-chaperoned SNARE assembly, no SNAP-25 binding and tomosyn SNARE assembly was observed when adding 60 nM SNAP-25 in solution (measured from N = 23 independent binding experiments, Fig. 4b, FEC#3). For comparison, the syb2/VAMP2-Syx conjugate forms the template complex in the force range of 3–7 pN with an extension change of 5.4 nm (Fig. 4b, FEC#4, blue dashed oval), which frequently bound to SNAP-25 in the solution to form the SNARE complex (the extension-drop indicated by the red arrow in Fig. 4b, #4). These results were confirmed by experiments at a constant mean force (Fig. 4e). In conclusion, the tomosyn SNARE motif alone failed to form a template complex with Munc18-1-bound open syntaxin-1. Hence, tomosyn cannot directly

compete with syb2/VAMP2 during Munc18-chaperoned SNARE assembly.

**Tomosyn binds to syb2/VAMP2-containing template complexes**
The above results show that tomosyns can only replace syb2/VAMP2 during spontaneous SNARE assembly, but not during Munc18-1-chaperoned SNARE assembly. To further explore the molecular mechanism underlying tomosyn's function, we examined whether the tomosyn SNARE motif modulates the formation of the syb2/VAMP2-containing template and SNARE complex. For this purpose, we used the syb2/VAMP2-Syx conjugate and added 2 μM Munc18-1 and 60 nM SNAP-25 in the solution without or with 2 μM tomosyn SNARE motif (Fig. 5a, b). As expected, Munc18 first reversibly bound to open syntaxin (Fig. 5b, FEC#1, state 6, blue rectangle), and subsequently to syb2/VAMP2 to form the template complex (state 7; blue oval). SNAP-25 can then bind the template complex resulting in fully assembled

neuronal SNARE complexes (state 1; red arrow). The different states and transitions are schematically depicted in Fig. 5c. When the tomosyn SNARE motif was further added to the solution (Fig. 5b, FEC#2), an intermediate state 7* frequently appeared in the otherwise binary transition between the open syntaxin state 6 and the template complex state 7 (schematically depicted in Fig. 5c, also compare the first and second trajectories in Fig. 5d), with an average extension ~1.5 nm greater than that of the template complex state and ~4 nm less than that of open syntaxin (Supplementary Fig. 8). This new state is syb2/VAMP2-dependent, as the tomosyn SNARE motif did not induce further folding of Munc18-1-bound open syntaxin-1 when either the tomosyn-syntaxin-1 conjugate (Fig. 4b, e) or syntaxin-1 alone (Fig. 5e and Supplementary Fig. 9) was being pulled. These observations suggest that the tomosyn SNARE motif specifically binds to the template complex to induce a large conformational change as represented by its 1.5 nm extension change. Hence, while the tomosyn SNARE motif is unable to replace syb2/VAMP2 during Munc18-1 chaperoned SNARE assembly, it can bind to the syb2/VAMP2-containing template complex.

We characterized the stability and folding kinetics of the tomosyn-bound template complex. To this end, we extensively measured the three-state transitions at different constant mean forces and analyzed the resulting time-dependent extension trajectories using hidden Markov modeling[88] (Fig. 5d). The analyses revealed the probabilities and lifetimes of the three states and their associated transition rates. Extrapolation of these force-dependent quantities to zero force using an energy landscape model of protein folding yielded the unfolding energies of the tomosyn-bound and unbound template complexes to be 3.1 ($\pm$0.1, S.E.) $k_BT$ and 5.1 ($\pm$0.2) $k_BT$, respectively (Fig. 5c)[90]. The unfolding energy of the template complex is consistent with our previous measurement of 5.2 ($\pm$0.2) $k_BT$[19], while the unfolding energy of open syntaxin (from state 6 to state 5) was previously measured to be 2.6 $k_BT$[19,84]. The average lifetimes of these states are ~0.7 s and ~1.6 s for the tomosyn-bound and unbound template complexes, respectively, as well as ~0.014 s for the Munc18-1-bound open syntaxin-1. These measurements demonstrate that tomosyn binding could destabilize the template complex. In the presence of 2 μM tomosyn SNARE motif in the solution, the tomosyn-bound template complex efficiently forms from two pathways, either coupled binding of tomosyn and syb2/VAMP2 to Munc18-1-bound open syntaxin-1 with a rate of ~ 28 s$^{-1}$ (from stage 6 to state 7*), or direct binding of tomosyn to the preformed template with a rate of ~1 s$^{-1}$ (from stage 7 to state 7*). In addition, the tomosyn SNARE motif dissociates from the template complex with a rate of 8 s$^{-1}$. Compared to the direct folding rate of 65 s$^{-1}$ for the template complex, the tomosyn-bound template complex is expected to compete with the template complex for its formation, thereby modulating Munc18-1-chaperoned SNARE assembly.

### The tomosyn-bound template complex does not bind SNAP-25

To uncover the function of the tomosyn-bound template complex in further SNARE assembly, we tested its binding to SNAP-25 during the three-state transitions at constant mean force in the presence of SNAP-25 in the solution. A distinct and irreversible extension decrease appeared, which only started from the extension corresponding to the template complex, but not the tomosyn-bound template complex (N = 28, Fig. 5b, FEC#3; Fig. 5d, bottom trace). Therefore, SNAP-25 only binds to the template complex (Fig. 5c, state 7)[19], but not the tomosyn-bound template complex (state 7*), to form the SNARE four-helix bundle. Due to the presence of the tomosyn-bound template complex, the probability to observe chaperoned SNARE assembly within ~100 s detection time was reduced to 0.13, compared to the probability of 0.7 in the absence of tomosyn[19]. Thus, tomosyn attenuates chaperoned SNARE assembly by binding to the template complex to inhibit its association with SNAP-25.

### The Tomosyn SNARE motif extensively binds to the template complex

To investigate the critical region for tomosyn binding to the template complex, we conducted a series of truncations within the tomosyn SNARE motif and assessed their interactions with the template complex (Fig. 6a, left panel). Removing the C-terminal region led to a reduction in the lifetime and probability of the tomosyn-bound template complex (Fig. 6a, compare trace ii to trace i in the right panel; Supplementary Fig. 11). Further truncation to the -4 layer of the SNARE motif abolished the tomosyn-bound template complex (Fig. 6a, trace iii). The C-terminal region of the tomosyn SNARE motif alone did not bind the template complex (trace iv). Furthermore, even the C-terminal polybasic domain of tomosyn enhanced binding, as its truncation reduced the lifetime and probability of tomosyn binding (Fig. 6a, trace v; Supplementary Fig. 10). Therefore, the majority of the SNARE motif contributes to tomosyn's binding to the template complex. Additionally, at the same concentration, free VAMP2 in the solution did not impair template complex or induce additional states (trace vi), reinforcing the cooperative role of tomosyn with VAMP2 in forming the tetrameric complex.

The large extension increase of the template complex (~1.5 nm) upon tomosyn binding could arise from either a significant conformational change in Munc18-1 or detachment of the C-termini of syntaxin and VAMP2 from Munc18-1 (Fig. 6b, left panel). To distinguish between these scenarios, we mutated the most C-terminal hydrophobic layers of syntaxin (I233G/E234G/Y235G at +2 layer) and VAMP2 (F77A at +6 layer) bound to Munc18-1 and tested their effects on tomosyn binding. As expected[19], both modifications significantly impair the template complex, as indicated by its lower probability and smaller extension change relative to the unfolded open syntaxin state 6 (Fig. 6b, compare traces i and iii with the first trace in Fig. 5d). Notably, tomosyn had minimal effect on the two-state transition (Fig. 6b, compare traces ii to i and traces iv to iii), which suggests that tomosyn does not strongly bind the template complex with an altered conformation. This observation rules out the scenario where tomosyn binding induces unfolding of the C-terminus of either SNAREs. Consequently, we lean toward the alternative hypothesis that tomosyn binding triggers a large conformational change in Munc18-1. Further experiments are needed to validate this hypothesis.

### Tomosyn's polybasic domain is critical

The above findings prompted us to re-evaluate the importance of the different functional domains in tomosyn, starting with the point mutation in the +6 layer of the SNARE motif (FA-mutant, Fig. 7a). This mutant was based on the F77A mutant in syb2/VAMP2 which abolished template complex formation, chaperoned SNARE assembly and secretory vesicle fusion[86,91,92]. In contrast to wild-type tomosyn, the FA-mutant failed to reverse the enhanced synaptic depression in cDKO neurons (Fig. 7b, c) or restore synaptic recovery after high-frequency stimulation (Fig. 7d, e). However, while the FA-mutant was targeted to synapses, its expression level was lower than wild-type tomosyn, which could partly contribute to its reduced rescue ability (Fig. 7f, g and Supplementary Fig. 11).

Tomosyns contain two N-terminal WD40 double propeller regions (together referred to as WD40 domains), a proposed auto-inhibitory tail domain[93-95], a SNARE motif and a polybasic domain at the COOH-terminal end that contains multiple lysines (Fig. 7a). We next tested a series of C-terminal truncations, stating with the C-terminal polybasic domain (ΔPB, Fig. 7a). The ΔPB-mutant failed to restore the cDKO phenotype (Fig. 7b–e), even though this mutant was expressed at similar levels as wild-type tomosyn (Fig. 7f, g and Supplementary Fig. 11). Two consecutive larger truncations either removed the SNARE motif (ΔSNARE-mutant; Fig. 7a), or the SNARE and tail domain, leaving only the WD40 domain intact (WDonly-mutant; Fig. 7a). Again, expression of

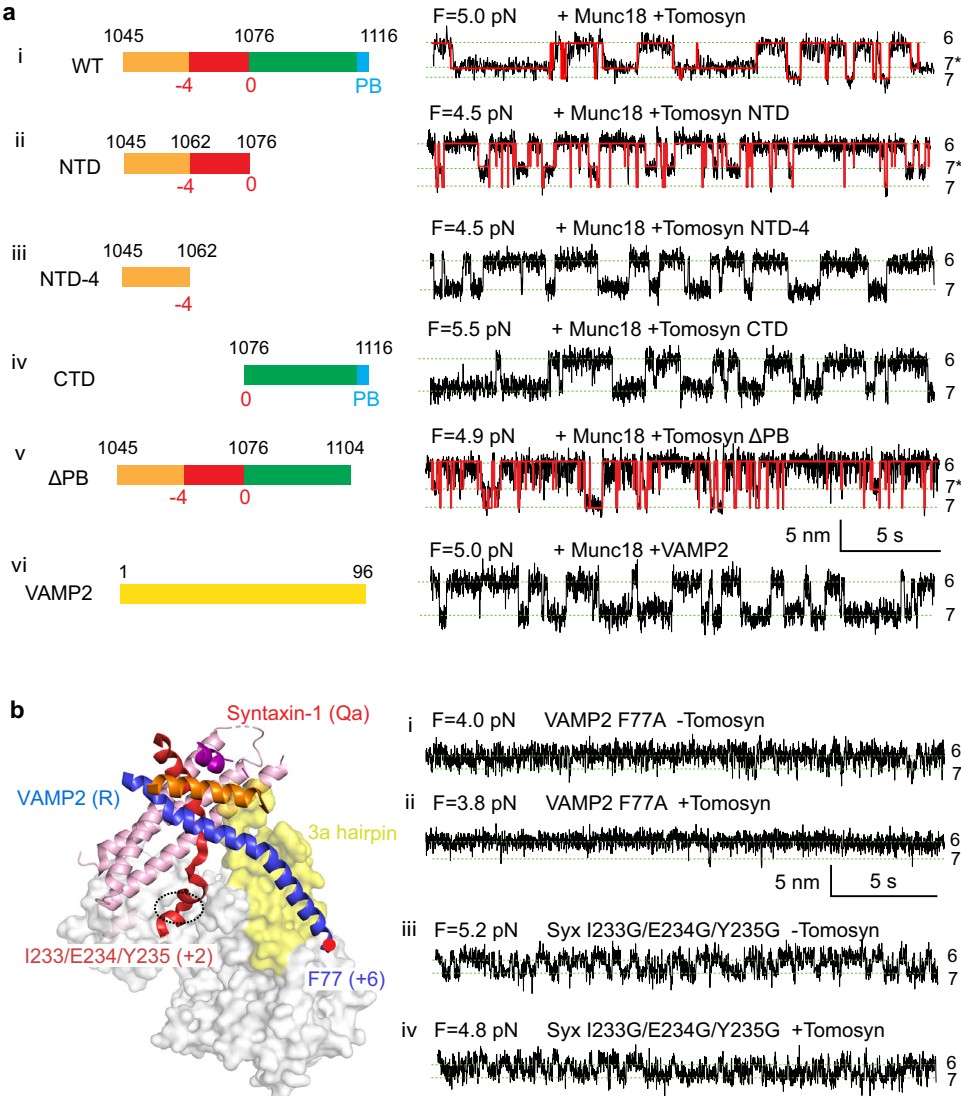

**Fig. 6 | The entire SNARE motif of tomosyn contributes to its binding to the template complex. a** Effects of tomosyn truncation or VAMP2 on the template complex. The left panel shows the schematic diagram of different tomosyn domains that remained in the truncated tomosyn relative to the entire SNARE domain, including the N-terminal SNARE motif or domain (NTD), the C-terminal SNARE domain (CTD), and the polybasic domain (PB). The right panel is the corresponding extension-time trajectory at constant mean force F in the presence of 2 μM free WT or truncated tomosyn or VAMP2 in the solution. The folding states

corresponding to different extensions are labeled on the right and illustrated in Fig. 5c. The red curves represent the idealized transitions derived from hidden Markov modeling. **b** Left panel: the structure of the template complex (PDB ID 1SFC) with the mutated C-terminal hydrophobic layers indicated. Right panel: the extension-time trajectories at constant mean force showing folding and unfolding transition of the templated complex in the absence and presence of the tomosyn SNARE motif. See also Supplementary Fig. 10.

both these mutants could not restore the cDKO phenotype (Fig. 7b–e), suggesting that the WD40 domains alone are not sufficient for tomosyn function, irrespective of the presence of the tail domain. Notably, just as the FA-mutant, these mutants were properly targeted to synapses but their expression levels were reduced compared to wild-type tomosyn (Fig. 7f, g and Supplementary Fig. 11). Hence, synaptic tomosyn protein levels might depend on the presence of a functional SNARE motif. All groups restored their initial EPSC size within 90 s after 10 Hz-induced depression (Supplementary Fig. 11f). Taken together, these results suggest that tomosyns require both an intact SNARE motif and C-terminal polybasic domain to function properly.

### The polybasic domain contributes to tomosyn vs syb2/VAMP2 diversity

The C-terminal region targeted in the tomosyn- syb2/VAMP2 hybrid tested above spanned both the SNARE motif and polybasic domain

(Fig. 3). To test whether one domain was solely responsible for the reduced functionality, partial hybrid constructs were designed: a core-hybrid in which only the SNARE motif was replaced and a linker region (LR)-hybrid in which the linker region of syb2/VAMP2 replaced the polybasic domain of tomosyn (Fig. 8a). These tomosyn variants were properly targeted to synapses, were expressed at similar levels and did not alter neuronal morphology (Fig. 8b and Supplementary Fig. 12). While wild-type tomosyn efficiently restored the defects in short-term plasticity and synaptic recovery in cDKO neurons, both the core-hybrid and LR-hybrid only partially rescued these synaptic defects (Fig. 8c–g), similar to our previous findings. All groups restored their initial EPSC size within 90 s after 10 Hz-induced depression (Supplementary Fig. 12f). Thus, the core SNARE motif and C-terminal polybasic domain both contributed to the reduced functionality of the original hybrid mutant, confirming the requirement of these domains for tomosyn function.

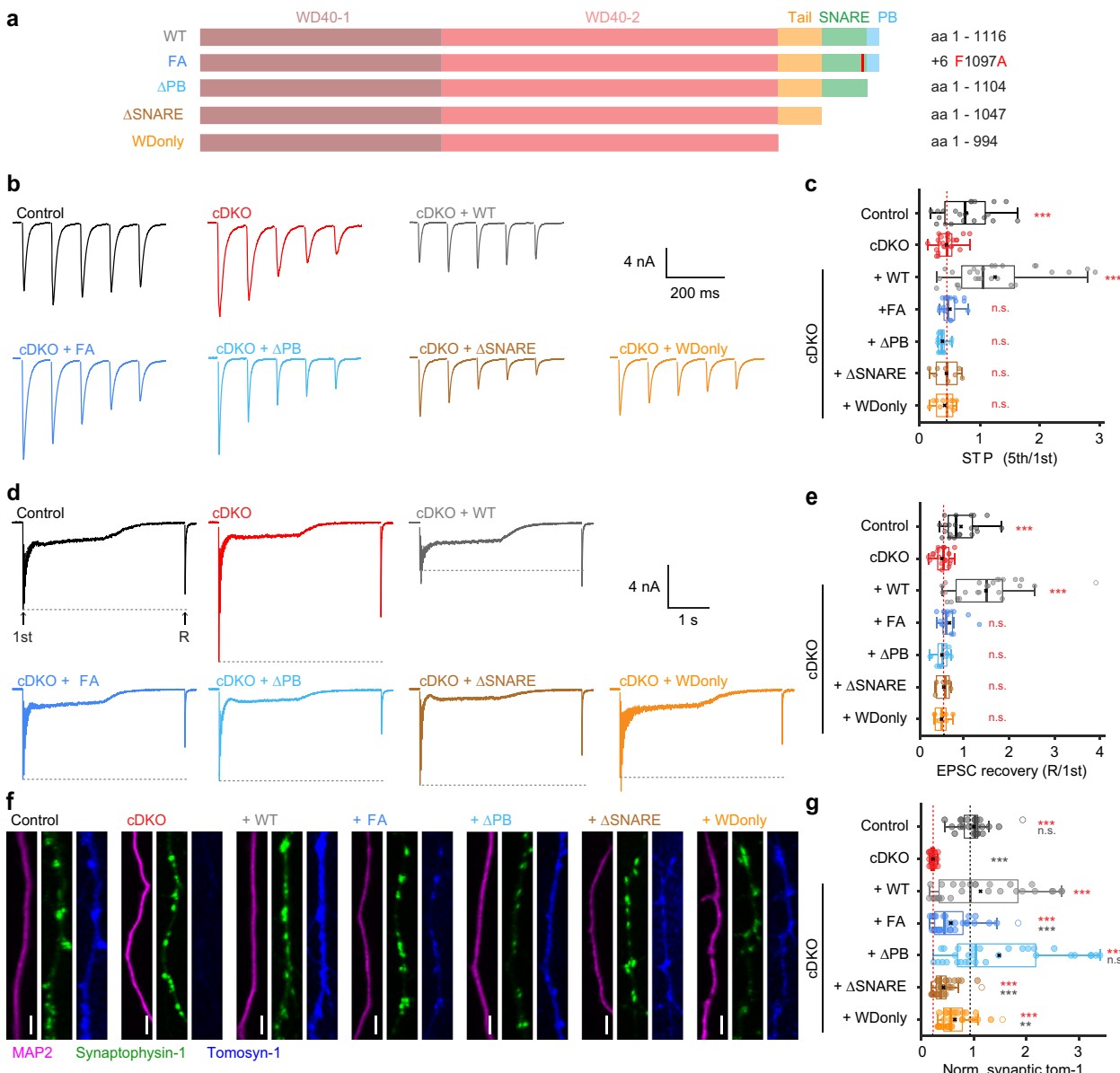

**Fig. 7 | A polybasic domain outside the SNARE motif is critical for tomosyns' function. a** Schematic representations of wild-type and mutant tomosyn-1m. The amino acid (aa) numbers and mutation site are stated on the right. **b** Example traces of 10 Hz. **c** STP quantified by the ratio of the fifth pulse over the first pulse. Control n = 21/6, cDKO n = 25/8, +WT n = 27/6, +FA n = 18/5, +ΔPB n = 13/4, +ΔSNARE n = 8/4, +WDonly n = 10/3. Post hoc comparisons against cDKO: ***p = 0.0001 (control); ***p < 0.0001 (+WT), p = 0.3013 (+FA), p = 0.8781 (+ΔPB), p = 0.7821 (+ΔSNARE), p = 0.2592 (+WDonly). **d–e** A recovery pulse (R) was given 2 s after 40 Hz stimulation. **d** Example traces. **e** Quantification of EPSC recovery (normalized to 1st). Control n = 21/6, cDKO n = 25/8, +WT n = 25/6, +FA n = 16/5, +ΔPB n = 12/4, +ΔSNARE n = 6/3, +WDonly n = 9/2. Post hoc comparisons against cDKO: ***p = 0.0001 (control); ***p < 0.0001 (+WT), p = 0.03677 (+FA), p = 0.4757 (+ΔPB), p = 0.6820 (+ΔSNARE), p = 0.8506 (+WDonly). **f, g** Immunostainings confirm expression of all mutants. **f** Example images of dendrites. Scale bar = 10 μm. **g** Synaptic tomosyn-1 expression normalized to control within each neuronal culture. Control n = 26/3,

cDKO n = 27/3, +WT n = 29/3, +FA n = 33/3, +ΔPB n = 30/3, +ΔSNARE n = 36/3, +WDonly n = 34/3. Post hoc comparisons against +WT: p = 0.1731 (control); ***p < 0.0001 (cDKO), ***p < 0.0001 (+FA), p = 0.0461 (+ΔPB), ***p < 0.0001 (+ΔSNARE), p = 0.0005 (+WDonly). Post hoc comparisons against cDKO: ***p < 0.0001 (control); ***p < 0.0001 (+WT), ***p < 0.0001 (+FA), ***p < 0.0001 (+ΔPB), ***p < 0.0001 (+ΔSNARE), ***p < 0.0001 (+WDonly). N = cells/independent cultures. In (**c, e, g**), boxplots display median (center), upper and lower quartiles (box bounds) and whiskers to the last datapoint within 1.5x interquartile range. A one-way ANOVA tested the significance of adding experimental group as a predictor, see Supplementary Table 1. For post-hoc comparisons, p-value thresholds (*<0.05; **<0.01;***<0.001) were adjusted with a Bonferroni correction. Red asterisks show comparison to cDKO, gray asterisks show comparison to + WT. Abbreviations: PB (polybasic domain), n.s. (not significant). See also Supplementary Fig. 11. Source data are provided as a Source Data file.

## Discussion

In this study, we characterized the role of the two mammalian tomosyn genes in SNARE complex assembly, SNARE-dependent membrane fusion and synaptic transmission. Hippocampal neurons lacking both tomosyns exhibited a decreased energy barrier for synaptic vesicle fusion and, consequently, an increased initial synaptic strength,

followed by more pronounced synaptic depression and slower recovery (Figs. 1, 2). While expression of wild-type tomosyn-1 fully rescued cDKO phenotypes, a hybrid containing the SNARE motif of syb2/VAMP2 did not (Fig. 3). In line with these findings, single molecule force measurements revealed that tomosyn's SNARE motif fails to form a tomosyn-template complex with Munc18-1 and syntaxin-1

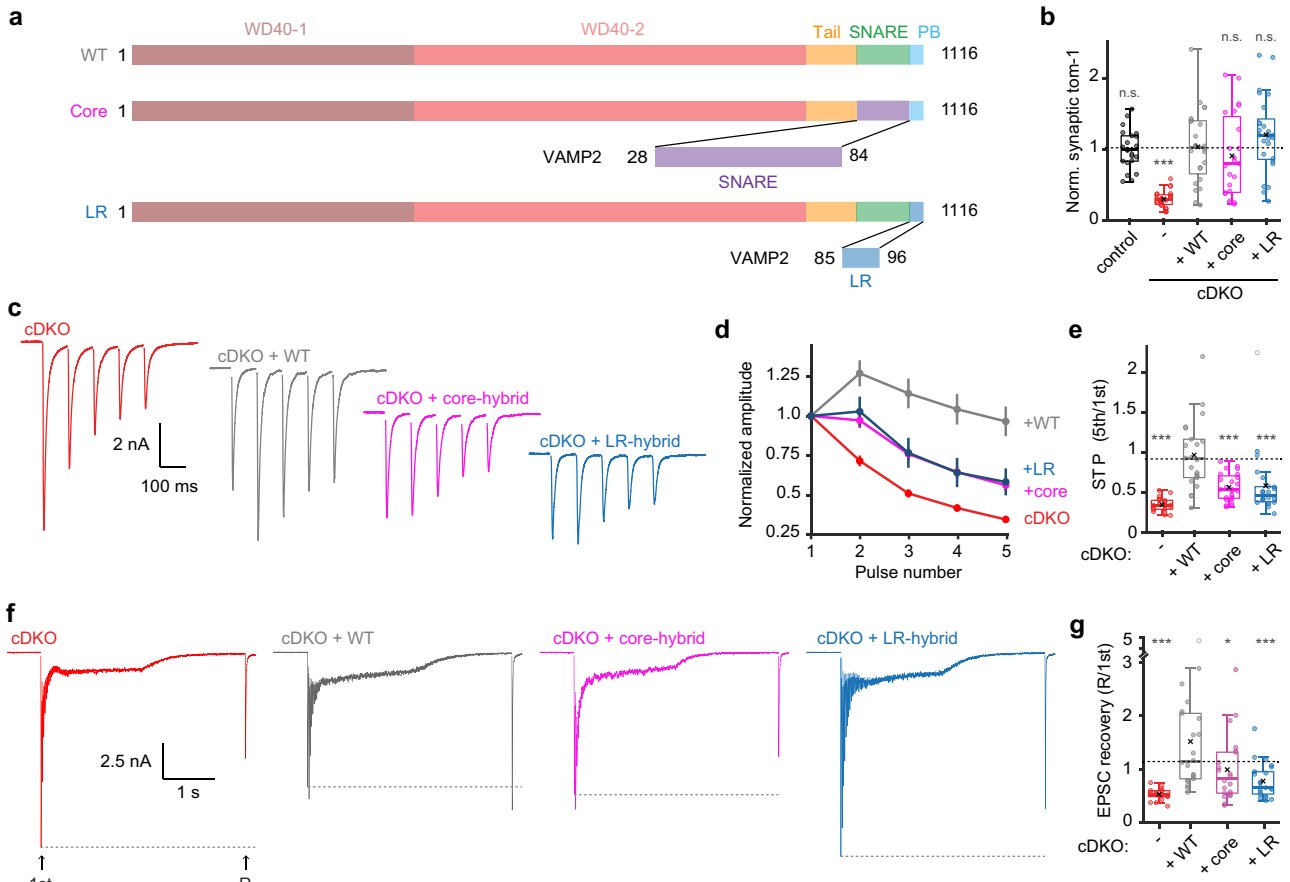

**Fig. 8 | The polybasic domain contributes to the reduced functionality of the VAMP2-hybrid. a** Schematic representation of mutant constructs created to test the individual contributions of the linker region and the SNARE motif to the lack of interchangeability of the corresponding regions in tomosyn and VAMP2. **b** Normalized synaptic tomosyn-1 expression. Control n = 22/4, cDKO n = 23/4, + WT n = 24/4, + Core n = 23/4, + LR n = 25/4. Post hoc comparisons against +WT: $p = 0.7549$ (control); ***$p < 0.0001$ (cDKO), $p = 0.3389$ (+Core), $p = 0.3419$ (+LR). See Supplementary Fig. 12 for example images and morphological analysis. **c–e** Short-term plasticity (STP) was tested by stimulation with 5 pulses at 10 Hz. cDKO n = 20/4, + WT n = 22/4, +Core n = 24/4, +LR n = 23/4. **c** Example traces. **d** Amplitudes were normalized to the first pulse. **e** STP is quantified by the ratio of the fifth over the first amplitude. Post hoc comparisons against +WT: ***$p < 0.0001$ (cDKO), ***$p < 0.0001$ (+Core), ***$p < 0.0001$ (+LR). **f, g** Recovery of the initial amplitude was tested by

high-frequency stimulation (80 pulses at 40 Hz) followed by a recovery pulse after 2 s. cDKO n = 19/4, +WT n = 22/4, +Core n = 22/4, +LR n = 22/4. **f** Example traces. **g** Quantification of EPSC recovery (normalized to 1st). Post hoc comparisons against +WT: ***$p < 0.0001$ (cDKO), *$p = 0.0043$ (+Core), ***$p < 0.0001$ (+LR). N = cells/independent cultures. In (**b**, **e**, **g**), boxplots display median (center), upper and lower quartiles (box bounds) and whiskers to the last datapoint within 1.5x inter-quartile range. In (**d**), data are presented as mean ± SEM. A one-way ANOVA tested the significance of adding experimental group as a predictor, see Supplementary Table 1. For post-hoc comparison to +WT group, $p$-value thresholds (*<0.05; **<0.01;***<0.001) were adjusted with a Bonferroni correction (α/number of tests). Abbreviations: PB (polybasic domain); LR (Linker region); n.s. (not significant). See also Supplementary Fig. 12. Source data are provided as a Source Data file.

(Fig. 4), but instead binds to the synaptic template complex VAMP2:Munc18-1:Syntaxin-1 to block its association with SNAP-25 (Fig. 5). Finally, the majority of tomosyns' C-terminus, including the polybasic region, contribute to this binding (Fig. 6) and is required for its effects on synaptic transmission (Figs. 7, 8).

Our data show that tomosyn deficiency leads to a substantially increased initial synaptic strength. This is consistent with previous findings in nematode and fly neuromuscular junction[39–41]. However, in these studies, tomosyn deficient synapses had larger docked and primed vesicle pools. In contrast, we found no evidence for enlargement of docked/primed pools. Mammalian and invertebrate neurons also show marked differences regarding the role of tomosyns in the other regulated secretion pathway (neuropeptide/neuromodulator release from dense core vesicles[96,97]).

Our results indicate that tomosyns control membrane fusion by increasing the energy barrier for fusion (Fig. 2), leading to decreased probability that vesicles fuse upon stimulation ($P_{ves}$). This is consistent with previous studies in the fly and mouse neuromuscular junction, where synapses lacking tomosyn had a high spontaneous release

frequency and exhibited faster synaptic depression[41,47]. Synaptic vesicles are known to have heterogeneous $P_{ves}$ (e.g., primed and super-primed vesicles)[16,98–102]. Tomosyns may shift the balance towards more low-$P_{ves}$ vesicles, causing the same stimulus to release a smaller fraction of the same pool of primed vesicles and thereby limiting initial synaptic strength, but also limiting pool depletion. Because tomosyns do not influence the vesicle recruitment/priming rate (Fig. 2k and Supplementary Fig. 5i), their net effect is to allow vesicle fusion to be more evenly distributed during action potential trains. These short-term plasticity phenomena are regarded as crucial in synaptic computation, such as working memory[103] and sensory processing[25].

Priming is defined as one or more molecular events upstream of the actual fusion reaction that brings vesicles in a readily releasable state and is widely considered to involve the initial interaction between SNARE proteins on the vesicle and active zone membranes. As a consequence, the energy barrier that needs to be overcome for the two lipid bilayers to merge is lowered. After this initial step, additional molecular events may further influence the remaining energy barrier, such as phorbol esters[72,75], complexins[104], Munc13 activation[74,105,106,

PKC-dependent phosphorylation of Munc18-1[8,107], synaptotagmin-1[108], or liprin-alpha[109], and synaptic plasticity phenomena such as augmentation[73] and post tetanic potentiation[80]. The effect of tomosyn deficiency on synaptic strength and short-term plasticity resembles the effect of these effects/phenomena and therefore likely acts at the same, post-priming step. We find that tomosyns bind the pre-assembled template complex, an early step downstream of the initial interaction between SNARE proteins (priming), to attenuate SNARE assembly and increase the fusion barrier (Fig. 9).

While tomosyns reduce synaptic strength at the onset of activity, synaptic responses following exhaustive stimulation were similar in control and tomosyn-deficient neurons (Fig. 2g–i). Under these conditions, most fusing vesicles are probably newly recruited vesicles. These vesicles may not yet be targeted by tomosyns And have a similar $P_{ves}$. This is further supported by the observation that tomosyns only increased the energy barrier for vesicles released during the initial sucrose application, but not in newly recruited vesicles that were released during a second application (Fig. 2l, m). Tomosyn-deficient neurons regained their initial enhanced EPSC size after short-term depression (Supplementary Figs. 7f, 11f, 12f) or after exhaustive stimulation (Fig. 2g–i) after 60-90 s recovery, suggesting that tomosyn-dependent suppression of $P_{ves}$ is fully reinstated within minutes. A recent paper suggests that stronger synapses generally recover more slowly from short-term depression due to a higher abundance of fully-primed but slowly-recovering vesicles[110]. Tomosyns may inhibit vesicles from entering such a high $P_{ves}$ state, effectively accelerating recovery from depression.

Our data argue against the concept that tomosyns directly compete with syb2/VAMP2 in SNARE assembly. This view is mainly derived from the observation that the tomosyn SNARE motif forms a stable ternary complex with syntaxin-1 and SNAP-25, similar to syb2/VAMP2[32]. We showed that, despite its much slower folding rate, the SDS-sensitive tomosyn SNARE complex is as thermodynamically stable as

the SDS-resistant synaptic SNARE complex (Fig. 4). Thus, our single molecule data generally support the previous observation on the stability of the tomosyn SNARE complex. However, two main findings challenge the in vivo relevance of this ternary complex: First, swapping the SNARE motif of tomosyn with that of syb2/VAMP2 failed to rescue tomosyn null mutant phenotypes, suggesting that the SNARE motif of tomosyns does not directly compete with syb2/VAMP2 for SNARE assembly (Fig. 3). Second, tomosyn's SNARE motif failed to form a template complex with Munc18-1 and syntaxin-1 in the way that syb2/VAMP2 does (Fig. 5). Consequently, Munc18-1 did not promote the assembly of tomosyn-containing SNARE complexes, consistent with a previous observation[31]. Taken together, our results challenge the proposed competition between tomosyns and VAMP2 during physiological SNARE complex assembly. Our data reveal that, instead, the SNARE motifs of both tomosyns and syb2/VAMP2 cooperatively bind to Munc18-1-bound open syntaxin-1 to form a tomosyn-bound template complex that blocks further SNARE assembly by association of SNAP-25 (Fig. 9a). Our finding is consistent with the negative role of tomosyn in SNARE assembly in the presence of both Munc18-1 and Munc13-1[31].

This study provides further evidence that physiological SNARE assembly is mediated by the template complex[17–19,21–23,111]. The template complex likely serves as a key target to regulate SNARE-mediated membrane fusion, for example via Munc13-1[112] and phosphorylation of Munc18-1[19]. The template complex is stabilized by extensive interactions between Munc18-1 and SNAREs[19,23], as well as interactions between the two aligned SNAREs. These interactions enhance the specificity of SNARE pairing. Thus, it is not surprising that tomosyns bind Munc18-1 in a way different from syb2/VAMP2, despite their similar SNARE motifs. In addition, both syb2/VAMP2 and syntaxin-1 are recruited by Munc13-1 and other regulatory proteins to form template complexes, and this recruitment depends on the membrane anchoring of both SNAREs[6,8,112]. These observations may explain why tomosyn

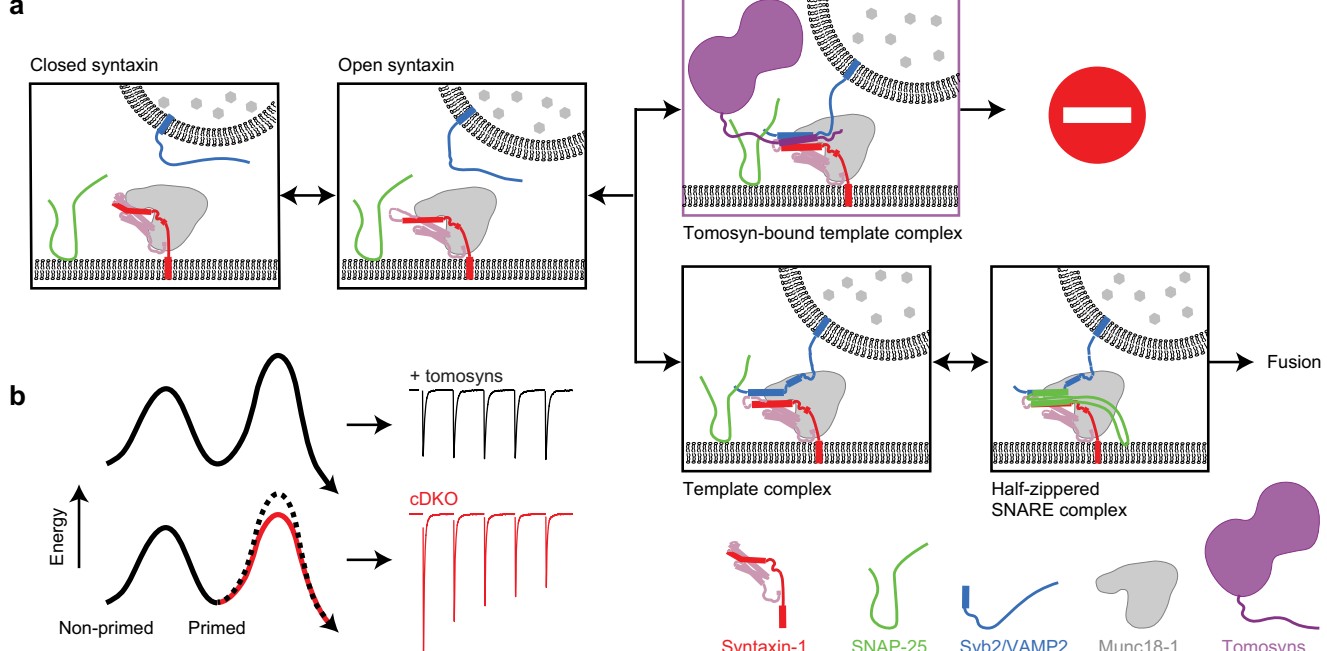

**Fig. 9 | Graphic representation of the working model of tomosyns attenuating SNARE assembly and synaptic depression. a** Munc18-1 sequesters syntaxin in a 'closed' conformation unable to associate with other SNAREs[118]. This closed dimer transits to an open conformation in which the SNARE motif is accessible, assisted by Munc13[119]. Syb2/VAMP2 binds to the open syntaxin complex to form the template complex in which the N-terminal regions of both SNARE motifs are aligned. Finally,

tomosyns bind the VAMP2-containing template complex, preventing SNAP-25 binding and further SNARE assembly. **b** Schematic of the energy landscape for synaptic vesicle priming and fusion. Tomosyns bind the assembling template complex to produce a higher fusion barrier SNARE complex state lacking SNAP-25. As a result, a smaller fraction of the vesicle pool will be released upon stimulation, which reduces synaptic strength and limits synaptic depression.

hybrids containing the SNARE motif of syb2/VAMP2 do not compete with vesicle-anchored syb2/VAMP2 for template complex formation (Figs. 3, 8). In conclusion, tomosyns target the syb2/VAMP2-containing template complex to regulate SNARE assembly and synaptic transmission.

In conclusion, our data show that tomosyns play a critical role in regulating synaptic strength, preventing primed synaptic vesicles from depletion during the onset of action potential trains (Fig. 9). In this way, tomosyns limit synaptic depression and stabilize information transfer during repetitive firing. The exact impact of this regulation for cognitive functions remains to be determined, but modeling of working memory and the association of tomosyn mutations with neurodevelopmental disorders both suggest that this impact is substantial. More experiments are required to elucidate the structure of the tomosyn-bound template complex and the impact of tomosyn's N-terminal domain on the structure.

## Methods

### Animals
Animal experiments were conducted under a CCD-protocol issued by the Central Authority for Scientific Procedures on Animals (CCD - Centrale Commissie Dierproeven) and approved by the Dutch government (VWA, Netherlands Food and Consumer Product Safety Authority) and local authorities (DEC, IvD VU-VUmc) in accordance with the European Council Directive (2010/63/EU). The generation of homozygous Tom2[lox] mice containing *LoxP* recombination sites flanking exon 3 of *Stxbp5l* was previously described[47]. A similar strategy was used to generate Tom1[lox] mice (Cyagen Biosciences). In short, a tomosyn-1 knock-in targeting vector was constructed by flanking exon 2 with *lox2272* recombination sites. Homologously targeted Tom1[lox] embryonic stem cells (C57BL/6) were injected into blastocysts and implanted into pseudopregnant females to produce germ-line chimeras, which were mated with inbred C57BL/6 mice. Tom1/2 double lox mice were obtained by mating C57Bl/6 Tom1[lox] mice with C57Bl/6 Tom2[lox] mice. Newborn pups from homozygous matings were used for the preparation of neuronal cultures in all the described experiments. Hippocampi from several pups from one nest were pulled in one culture preparation, which was considered as one biological replicate. Information on sex was not collected.

### Cell culture
Animals were sacrificed on postnatal day 1 (P1). Hippocampi were dissected in cold Hanks' balanced salt solution (Sigma, H9394) supplemented with 10 mM HEPES (Gibco, 15630-056) and digested with 0.25% Trypsin (Gibco, 15090-046) for 10–20 min at 37 °C. After washing three times with Hanks-HEPES the hippocampi were triturated using fire-polished glass pipettes in DMEM-glutamax (Gibco, 31966-021) supplemented with 10% heat-inactivated fetal calf serum (Gibco, 10270), 1% non-essential amino acids (Sigma, M7145) and 1% Penicillin-Streptomycin (Gibco, 15140-122). For functional analyses, neurons were plated on 18 mm glass coverslips in 12-well plates at a density of 2000 cells/well for single (autaptic) cultures and 8000 cells/well for micro-network cultures. Coverslips contained micro-islands of freshly prepped rat glia (see details below) in neurobasal medium (Gibco, 21103-049) supplemented with 2% B-27 (Gibco, 17504-044), 1.8% HEPES, 0.25% glutamax (Gibco, 35050-038) and 0.1% Penicillin-Streptomycin. Micro-islands were generated as described previously[61]. Briefly, etched glass coverslips coated with agarose (Type II-A; Sigma, A9918) were stamped with 0.1 mg/mL poly-D-lysine (Sigma, P6407) mixed with 0.7 mg/mL rat tail collagen (BD Biosciences, 354236) and 10 mM acetic acid (Sigma, 45731) using a custom-made stamp. Rat glia were prepared from the cortices of newborn rats (Wistar, strain code 003). Cortices were digested in papain (Worthington Biochemical Corporation LS003127) at 37 °C for 45 min and triturated in supplemented DMEM. Glia were plated and expanded in

T175 flasks. Freshly cultured rat glia were plated at a density of 8000/well for 5 days until confluent. For high density cultures for Western blot, neurons were plated at 300k/well on poly-L-Ornithine (Sigma, P4957) and laminin (Sigma, L2020) coated 6-well plates. Neurons were infected with lentiviral particles on day in vitro (DIV) 3 or 6 and used for experiments on DIV 13-18.

### Plasmids
Plasmids for lentiviral expression of EGFP-tagged active or inactive cre-recombinase behind a synapsin promotor were obtained from Pascal Kaeser[59,113] and possess an additional nuclear localization sequence of nucleoplasmin[114] in the N-terminus of EGFP to ensure optimal nuclear localization. Rescue constructs encoding full-length tomosyn-1 (NP_001074813.2, aa1-1116) and truncated and mutated versions contained a N-terminal mScarlet via a T2A sequence for selection of infected neurons and were cloned into a synapsin promoter-driven construct, sequence verified and subcloned into pLenti vectors. Viral particles were produced using HEK cells[115].

For proteins used in single molecule study, the amino acid sequences corresponding to WT SNARE and Munc18-1 were described elsewhere in detail[19]. The genes containing the cytoplasmic domains of rat syntaxin-1(aa1-265), syb2/VAMP2(aa1-96), SNAP-25(aa1-206), mouse tomosyn-1 R-SNARE(aa1045-1116) and rat Munc18-1(aa1-594) were cloned into the pET-SUMO vector encoding 6xHis-tag followed by a SUMO tag at the N termini. The full-length cysteine-free SNAP-25 was cloned into pET-15b vector encoding 6xHis-tag at the N terminus.

### Protein expression, purification and SNARE preparation
All SNARE proteins and Munc18-1 were expressed in BL21 E. coli cells at 37 °C for 3 h with 1 mM isopropyl β-D-1-thiogalactopyranoside (IPTG). Proteins then were purified using Ni-NTA-agarose beads and eluted with 300 mM imidazole and exchanged to buffer containing 25 mM HEPES (pH 7.4), 140 mM KCl, and 2 mM tris(2-carboxyethyl) phosphine (TCEP). Syntaxin-1 was then biotinylated at its C-terminal Avi-tag with the biotin ligase BirA. Ternary SNARE complexes were prepared and crosslinked with DNA handles as was previously described[19]. Briefly, syntaxin-1, SNAP-25, and syb2/VAMP2 or tomosyn were mixed at a molar ratio of 0.8:1:1.2, incubated at 4 °C, and purified using the 6xHis-tag on SNAP-25 and Ni-NTA-agarose. The eluted SNARE complexes were crosslinked with DTDP (2,2'-dithiodipyridine disulfide) treated DNA handles with a molar ratio of 50:1 in 100 mM phosphate buffer, 500 mM NaCl, pH 8.5.

### Dual-trap optical tweezers
The optical tweezers were home-built as described elsewhere. Briefly, a 1064 nm laser beam is expanded, collimated, and split into two orthogonally polarized beams, one of which is reflected by a mirror attached to a nano-positioning stage (Mad-city Labs, WI). The two beams are then combined, expanded again, and are then focused by a water-immersed 60x objective with a 1.2 numerical aperture (Olympus, PA) to form two optical traps in the sample plane in the central channel of a home-built microfluidic flow chamber. One of the two traps is stationary; the other trap can be moved using the nano-positioning stage. The outgoing laser beams are collimated by a second water-immersed objective, split again by polarization, and projected onto two position-sensitive detectors (Pacific Silicone Sensor, CA). Displacements of the trapped beads are detected by back-focal plane interferometry. Optical tweezers are remotely operated through a computer interface written in LabVIEW (National Instruments, TX). The force constants of two optical traps are determined by the Brownian motion of the trapped beads before each experiment.

### Single-molecule experiments and data analysis
An aliquot of the crosslinked protein-DNA sample was incubated with anti-digoxigenin coated polystyrene beads 2.17 μm in diameter

(Spherotech, IL), diluted in phosphate-buffered saline (PBS), and injected into the top channel of a microfluidic chamber. Streptavidin-coated polystyrene beads of 1.76 μm were injected into the bottom channel. Both top and bottom channels were connected to a central channel by capillary tubes, where both kinds of beads were trapped. A single SNARE complex was tethered between two beads by bringing them close. Data were recorded at 20 kHz, mean-filtered to 10 kHz. The single-molecule experiment was conducted in PBS at 23(±1) °C. An oxygen scavenging system was added to prevent potential protein photo-damage by optical traps. The single protein-DNA tether was pulled or relaxed by increasing or decreasing trap separation at a speed of 10 nm/s.

The methods of data analysis were described in detail elsewhere[88–90]. Briefly, by analyzing the extension trajectories using a two- or three-state hidden-Markov modeling (HMM), the probability, extension, force, lifetime (Supplementary Fig. 10), and transition rates for each state were obtained. To relate the experimental measurements to the conformations and energy (or the energy landscape) of different SNARE states at zero force, we constructed structural models for these states based on crystal structures of the SNARE four-helix bundle and the template complex. These states were characterized by the contour lengths of the unfolded polypeptides and free energy, which were chosen as fitting parameters. The extension and energy of the whole tethered dumbbell, including the DNA handle, were calculated using the Marko-Siggia formula. Then, we computed the probability of each state based on the Boltzmann distribution and transition rates based on the Kramers' equation. Finally, we fit the calculated state extensions, forces, probabilities, and transition rates to the corresponding experimental measurements using the nonlinear least-squares fitting, which revealed the conformations and energies of different SNARE folding states as best-fit parameters.

## Western blot

High density neuronal cultures were lysed on DIV14 in Laemmli sample buffer. Lysates were heated at 95 °C for 5 min and loaded on sodium dodecyl sulfate (SDS) polyacrylamide gels. Proteins were resolved at 100 V and transferred to nitrocellulose membranes using wet tank transfer system (Bio-Rad). Successful transfer was validated by Ponceau S staining of membranes. Membranes were blocked for 30 min in 5% milk powder dissolved in Tris-buffered saline containing 0.1% Tween-20 detergent (TBST, pH 7.4). Thereafter, membranes were incubated overnight at 4 °C with primary antibodies dissolved in blocking buffer. The following primary antibodies were used: polyclonal rabbit anti-tomosyn (1:1000, Synaptic Systems 183103), monoclonal mouse anti-syntaxin (1:2000, Sigma S0664), monoclonal mouse anti-SNAP-25 (1:1000, Covance SMI-81R), monoclonal mouse anti-VAMP2 (1:1000, Synaptic Systems 104211), monoclonal mouse anti-actin (1:2000, Chemicon MAB1501). After the incubation with primary antibody solutions, membranes were washed three times in TBST and incubated for 1 h at room temperature with horseradish peroxidase coupled secondary antibodies (Agilent Dako) diluted at 1:10000 in blocking buffer. Membranes were rinsed three times in TBST and developed on the Odyssey Fc imaging system (LI-COR Bioscience) using SuperSignal West Femto Maximum Sensitivity Substrate (Thermo Scientific). Chemiluminescent signals were analyzed using Image Studio Lite Software. Bands for tomosyn-1 were corrected for loading by normalizing to the corresponding actin bands. Then, the percentage of control was calculated using the corrected values. Four biological replicates were used. For each biological replicate 3 technical replicates were blotted.

## Immunocytochemistry

Neurons were fixed with 3.7% formaldehyde (Electron Microscopy Sciences) on DIV 15. After 20 min, cells were washed with home-made phosphate-buffered saline (PBS) and stored or directly permeated using 0.5% Triton X-100 for 5 min. Blocking solution (BS) contained 0.2% normal goat serum and 0.1 % Triton X-100. BS was applied for 30 min followed by 2 h primary antibody incubation at room temperature (RT). Antibodies were diluted in BS. Cells were stained for MAP2 as a dendrite marker and synaptophysin-1 as a synapse marker. The following antibodies were used: polyclonal chicken anti-MAP2 (1:500, Abcam ab5392), polyclonal guinea pig anti-synaptophysin-1 (1:500, Synaptic Systems 101004), polyclonal rabbit anti-tomosyn (1:500, Synaptic Systems 183103). After washing three times with PBS, cells were incubated with secondary antibodies diluted 1:1000 in BS for 1 h. The following secondary antibodies were used: goat anti-chicken Alexa546, goat anti-guinea pig Alexa488, goat anti-rabbit Alexa647 (Molecular Probes). After another three washes in PBS, coverslips were mounted on microscope slides using DABCO-Mowiol (Invitrogen). Cells were imaged on a confocal laser-scanning microscope (Nikon Eclipse Ti A1) using a 40× oil immersion objective (NA 1.3). Only EGFP-positive autapses were imaged. Neuronal morphology, synapse numbers and the average tomosyn intensity per neuron were analyzed using the semi-automated platform SynD[65] (www.johanneshjorth.se/SynD). For each morphological parameter a mask was created based on the marker signal. The neurite mask was based on MAP2 signal. The synapse mask was created based on synaptophysin-1 puncta. Average tomosyn levels were measured within these puncta.

## Electrophysiology

Autaptic neurons were subjected to whole-cell voltage-clamp recordings (Vm = −70 mV) on DIV 13-16. Experiments were performed at room temperature with borosilicate glass pipettes (Science products GmbH, 2.5-4.5 Mohm) filled with (in mM): 136 KCl, 17.8 HEPES, 1 EGTA, 0.6 MgCl2, 4 ATP-Mg (Sigma, A9187), 0.3 GTP-Na (Sigma, G8877), 12 phosphocreatine dipotassium salt (Calbiochem, 237911) and 50 U/ml phosphocreatine kinase (Calbiochem, 2384) (pH = 7.3, ~300 mOsmol). Extracellular solution contained the following (in mM): 10 HEPES, 10 Glucose, 140 NaCl, 2.4 KCl, 4 MgCl$_2$ and 2 CaCl$_2$ (pH = 7.30, ~300 mOsmol). Solutions were made from stock with HEPES and Glucose added fresh, solutions were filter-sterilized and stored at 4 °C until use. Patch-clamp recordings were performed with a MultiClamp 700B amplifier and Digidata 1550B or an Axopatch 200B amplifier and Digidata 1440 A, controlled by Clampex 10.6 software (Molecular Devices). For gap-free recordings of spontaneous miniature EPSCs, the sampling rate was set to 20 kHz and low-pass Bessel filter was set to 5–6 kHz. For episodic stimulations, the sampling rate was set to 10 kHz and low-pass Bessel filter to 2 kHz. Resistance was compensated by 70–80% (bandwidth 7.52 Hz). Feedback resistor 500 MΩ was adjusted only if EPSCs were larger than 20 nA. Action potentials were elicited by a 1 ms depolarization to 30 mV. Recordings were excluded if series resistance was higher than 15 MΩ, leak current exceeded 300 pA or EPSC size was below 300 pA. GABAergic recordings were identified based on their postsynaptic decay kinetics and excluded. Offline analysis was performed with MATLAB R2018b and R2019a (Mathworks) using custom-written software routines (github.com/vhuson/view-EPSC). EPSC kinetics were calculated in Clampfit 10.6.

## Micro-networks

Micro-networks consisting of 3–10 neurons (DIV14-18) were recorded and analyzed as described above with the following modifications. The GABA$_A$ receptor antagonist picrotoxin (PTX, 50 μM, Hellobio) was added to the extracellular solution to block inhibitory currents. Spontaneous vesicle release was measured in the presence of 1 μM tetrodotoxin (TTX). Action-potential evoked EPSCs were elicited by local electric stimulation (1.5 mA, 1 ms) using a concentric electrode (CBCBG75, FHC). For evoked release, intracellular solution contained (in mM): 115 Cs-gluconate, 10 HEPES, 10 TEA-OH, 1 EGTA, 4 CsCl, 2 ATP-Mg, 0.4 GTP-Na$_2$, 10 phosphocreatine, 1 QX314-Cl (pH = 7.30).

## Hypertonic sucrose application

250 mM and 500 mM sucrose (Sigma-Aldrich) solutions were freshly made from aCSF at the start of each experimental week. Sucrose was applied via gravity infusion through a custom-made barrel system controlled by a perfusion Fast-Step delivery system (SF-77B, Warner instrument corporation). Speed of flow was controlled with an Exadrop precision flow rate regulator (B Braun). Multiple sucrose applications were performed in the following order: 250 mM, 3 min recovery time, 500 mM, 30 s recovery time, 500 mM. Each application of sucrose lasted 7 s. In between sucrose applications, cells were constantly perfused with aCSF. In manual analysis, the charge released within the first 4 s of sucrose application was calculated. Alternatively, the sucrose traces were fitted with a minimal vesicle state model as published previously[75,80] using MATLAB code kindly shared by Vincent Huson. This method yields a more accurate measure of the readily releasable pool (RRP) compared to other methods by accounting for ongoing priming. Additional measures obtained from this fitting are priming and de-priming rates, the fusion rate (k2) and the energy barrier for fusion ($\Delta Ea$). The release rate and energy barrier at 0 mM sucrose were obtained by analyzing the mEPSCs within the first 6 s before the first application of sucrose, recorded in the same traces. The frequency of miniature EPSCs (mEPSCs) within this time was multiplied by the average charge of one mEPSC; this value was then divided by the RRP measure to calculate k2 at 0 mM sucrose. An estimation of the energy barrier results from computing the natural logarithm of k2 (see in ref. 75 for method details). The average estimation from WT cells was used to calculate the change in the energy barrier evoked by applications of sucrose. The time of sucrose onset was analyzed using Clampfit 10.6 (Molecular Devices) by documenting the position of a cursor manually placed at the start of a sucrose response.

## Electron Microscopy

Dissociated hippocampal mouse neurons (5 k/well) were plated on pre-grown cultures of rat glia on sapphire disks (Wohlwend GmbH) to form micro-networks of 2–10 neurons per sapphire disk. Prior to culturing, sapphire disks were etched for 30 min in 60% sulfuric acid, washed, incubated in 3 M KOH overnight, washed and dried before carbon coating and subsequent baking at 180C for 2 h. Sapphire disks were coated by a mixture of 0.1 mg/ml poly-d-lysine (Sigma), 0.7 mg/ml rat tail collagen (BD Biosciences), and 10 mM acetic acid (Sigma) and placed in an agarose-coated 12-well plate to form glia monolayer islands selectively on sapphire disks. The sapphire disks were cryofixed on DIV14 in an EM-PACT2 (Leica Microsystems) high-pressure freezer in 5% trehalose/10% BSA in 0.05 M phosphate buffer (pH 7.4, 320 mOsm) cryoprotectant. Frozen samples were postfixed in 1% OsO4/ 5% saturated K4Fe(CN)6 in H2O in acetone at −90 °C for 74 h and brought to 0 °C at 5°/h. After several washes with ice-cold acetone, the sapphire disks were washed with propylene oxide and infiltrated by an increasing EPON concentration series. The samples were embedded in fresh EPON overnight and left to polymerize at 65 °C for 48 h.

Sapphire disks were separated from the EPON by dipping the samples in boiling water and liquid nitrogen and regions with neuronal networks were selected by light microscopy. These regions were cut out and mounted on pre-polymerized EPON blocks for ultrathin sectioning. Ultrathin sections (80 nm) were cut parallel to the sapphire disk, collected on single-slot, formvar-coated copper grids, and stained in uranyl acetate and lead citrate. Hippocampal synapses were randomly selected at low magnification using an electron microscope (JEOL1010) at 60 kV while being blinded for the experimental conditions. The number of docked SVs, total SV number, and active zone length were measured on digital images taken at 80,000-fold magnification using custom-written semiautomatic image analysis software running in Matlab (Mathworks). The distribution of active zone proximal vesicles was measured in synapse profiles as the shortest distance from the vesicular membrane to the plasma membrane within the active zone region and plotted in 5–10 nm bins using the same software. For all morphological analyses the following requirements were set: clearly recognizable synapses with intact synaptic plasma membranes with a recognizable pre- and postsynaptic area and defined SV membranes. SVs were defined as docked if there was no distance visible (<0.7 nm) between the SV membrane and the active zone plasma membrane.

## Statistics

Datasets on single neuron parameters consist of several neuronal cultures (N = number of independent cultures), in which different coverslips from the same culture are infected with different viruses resulting in separate experimental groups (e.g., control and cDKO correspond to deltaCre-EGFP and Cre-EGFP infected neurons), from which multiple observations (n = individual neurons) are taken. To account for the nested nature of our data, we performed fixed linear regression in which culture was included as a linear predictor. Outliers, defined as datapoints more than 3 standard deviations above or below the group mean, were removed. Data were then standardized into Z-scores by grand mean centering. A fixed linear regression model was fitted to the standardized data using the lm() function in R (version 4.1.0). A one-way anova (analysis of variance) was used to assess whether including the experimental group as a second linear predictor (formula = y ~ Group + Culture) statistically improved the fit of a model without group information (formula = y ~ 1 + Culture). When more than two experimental groups were present, post-hoc analysis was performed repeating the fixed linear regression on pair wise subsets of the data, and p-value thresholds were Bonferroni-adjusted to account for multiple testing.

Electron microscopy data on low density networks was tested by multilevel analysis using the lme() function in R (version 4.1.0) with sapphire disc as nesting level with the highest intracluster correlation of the data. A one-way anova was used to assess whether including the experimental group as a predicted variable (formula = y ~ Group, random = ~1|Sapphire, method = "ML") statistically improved the fit of a model without group information (formula = y ~ 1, random = ~1|Sapphire, method = "ML"). Post hoc pairwise contrasts were extracted by estimated marginal means (emmeans() function), and p-value thresholds were Bonferroni-adjusted to account for multiple testing.

Descriptive statistics are reported (mean, SEM, n and statistical details) in the Supplementary Table 1, and as box plots unless stated otherwise. On each box, the center line marks the median, a cross marks the mean, the box limits indicate the upper and lower quartiles and the whiskers extend to the last datapoint within 1.5× interquartile range. Individual datapoints are depicted as filled circles, with open circles representing outliers.

### Reporting summary

Further information on research design is available in the Nature Portfolio Reporting Summary linked to this article.

## Data availability

All data needed to evaluate the conclusions in the paper are presented in the main text and Supplementary Information file. Other data and material are available from the corresponding authors upon request. Source data are provided with this paper.

## Code availability

Published code from the semi-automatic image analysis platform SynD is available on Github [https://github.com/Hjorthmedh/SynD][116]. Custom code to analyze electrophysiological data is available on Github [https://github.com/vhuson/viewEPSC][117].

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

## Acknowledgements

We would like to thank Joke Wortel for animal breeding, Ingrid Saarloos for cloning, Robbert Zalm for producing viral particles, Lisa Laan and Desiree Schut for preparing glia cultures, and Joost Hoetjes for genotyping. We acknowledge Rien Dekker for high-pressure freeze electron microscopy. Furthermore, we would like to thank Vincent Huson for providing us with code and for assistance with fitting of hypertonic sucrose traces. We thank Niels Cornelisse, Ruud Toonen and Jacob Sørensen for their helpful discussion and comments on this work. This work is supported by the ZonMw-Veni program (09150161810052 to M.M.) from the Dutch Research Council (NWO), the ERC Advanced Grant (322966 to M.V.) of the European Union, the NWO Gravitation program grant BRAINSCAPES (NWO 024.004.012 to M.V.), the Horizon 2020 grant COSYN (RIA grant agreement no 610307, to M.V.), the Lundbeck Foundation Grant (R277-2018-802 to M.V.), the DFG (German Research Foundation) postdoctoral fellowship (DFG project number SU 1131/1-1 to A.S.) and the NIH grant R35 GM131714 to Y.Z.

## Author contributions

M.M, M.Ö., J.Y., Y.Z. and M.V. conceptualized the study. M.M. and M.Ö. conducted and analyzed electrophysiology and immunofluorescence experiments in single neurons. M.M. and T.V. conducted and analyzed electrophysiology in micro-network cultures. J.Y., A.K., Z.F. and Y.Z. performed single molecule force measurements and analyzed the data, A.S. performed and analyzed Western blots. J.v.v.W. supervised and analyzed electron microscopy experiments. A.J.G. designed conditional tomosyn mice and performed initial characterizations. M.M., M.Ö., J.Y., Y.Z. and M.V. wrote the manuscript with input from all other authors.

## Competing interests

The authors declare no competing interests.
