## [Peer Review File · Nature Communications]

Tomosyns attenuate SNARE assembly and synaptic depression by binding to VAMP2-containing template complexesREVIEWER COMMENTS

Reviewer #1 (Remarks to the Author):

Tomosyn has long been reported to be a negative regulator of neuronal SNARE complex assembly and synaptic secretion. The underlying mechanism of which is suspected to be that Tomosyn competes with VAMP2 to form a non-fusogenic SNARE complex with Syx1 and SNAP25. In this manuscript, Meijer et al. derived novel mouse model with Tomosyn-1/2 conditional deficit and found that Tomosyn retains synaptic recovery during short-term repetitive stimulation. The data further showed that Tomosyn binds to the Munc18-1/VAMP2/Syx1 template complex and prevents the entry of SNAP25, which is distinct to the canonical model. The manuscript would add a new layer to our understanding of the molecular mechanism of Tomosyn in regulating synaptic transmission. However, there remains some critical issues as well. I am favor of publishing the results if the author could address the concerns raised below properly.

Major comments:

1. It is clear that the pair-pulse ratio (PPR) of Tomosyn-1/2 cDKO is lower than the control, which may denote that synapses depress faster in the absence of Tomosyns. However, the absolute EPSC amplitude and charge transfer of the second pulse (even the fourth pulse) of cDKO are larger than those of Control (Fig.1h-j). I am not firmly sure whether comparing the absolute values would make sense in such experiments. However, from a non-professional point of view, the synapse strength seems not decrease (i.e., the absolute amplitudes of cDKO are not 'decreased' compared to Control, Fig.1h, i), instead, it only decreases from a much higher level (i.e., hyperactive) to a 'normal' level. Please comment on the issues.
2. In pg.3, lines 40-42, the authors wrote that 'Thus, tomosyns lower the release probability of synapses, thereby reducing their initial strength and limiting synaptic depression.' According to the early study on pair-pulse facilitation [Dobrunz and Stevens, 1997, PMID: 9208866], it is apparent that pair-pulse facilitation inversely correlates with release probability of pulse 1. The data presented in Fig.1f and g shows that the PPR of cDKO is lower than that of Control. The 'pair-pulse facilitation' in the above citation (also cited by the authors in the manuscript) is different to the 'pair-pulse ratio (PPR)' assessed in the manuscript, where the former is the ratio of release probability for the second pulse to the first pulse. Could these be equivalent? Please clarify.
3. In Fig.1, the authors adopted 5 Hz stimuli, whereas 10 Hz stimuli was applied in Fig.3d, Fig. 6b, and Fig.7c. The authors termed them all as the 'STP'; meanwhile, the 'depression' degrees seem to be different when using different frequency. Please comment on the issue or present reasonable explanations.
4. In pg.5, lines 28-29, 'Expressing tomosyn-1m... did not affect SV docking in cDKO cultures (+WT), but the number of docked SVs was reduced compared to control neurons.' It is unclear what the authors will express here. Please check the interpretation.
5. The data in Fig.2g-i illustrated that Tomosyn is responsible for short-term (2 s) instead of long-term (1 min) synaptic recovery, according to the authors' interpretation. However, the absolute EPSC amplitudes between cDKO and Control in short-term recovery displayed no significant difference. On its face, does it represent that Tomosyn has no role in this short-term recovery pulse? Or can we assume that the short-term recovery (i.e., <2 s, or maybe longer) utilizes different mechanism that does not require the

participation of Tomosyn and, if possible, Munc18s and/or other regulatory proteins? Certainly, the hypothesis is based on the premise that the augmented EPSC amplitude of the first pulse by Tomosyn-cDKO is abnormal (decrease the energy barrier of SV fusion as indicated by the authors). The authors seemed to present an arbitrary conclusion and did not conduct enough interpretations and discussions.

6. It is interesting that Tomosyn would hinder the entry of SNAP25 into the template complex. Although the authors found that addition of Tomosyn induces a conformational change (1.5 nm greater than state 7) of the template complex through OT measurements, the interaction pattern between Tomosyn and the template complex is still quite unimaginable. One possible mechanism may be that Tomosyn finitely competes the binding of VAMP2 to Munc18, since the C-terminal half of the SNARE motif of Tomosyn shares considerable homology with that of VAMP2. Meanwhile, the data in Fig. 6 (FA variant) also corroborate this. The authors should at least test the action of Tomosyn FA variant, Δ PB variants, Core-hybrid, LR-hybrid, and if possible, free VAMP2 C-half (or chimeric VAMP2-Tomosyn, or equivalents) in OT measurements to confirm the possible mechanism.

7. It is clear that the poly-basic tail of Tomosyn play an important role in regulating neuronal SNARE complex assembly as shown in Fig.6. The authors concluded in the Discussion section that ‘..the SNARE motif and the polybasic domain of tomosyns might cooperatively interfere with C-terminal zippering of partially assembled SNARE complexes into the juxtamembrane domains of syb2/VAMP2 and syntaxin-1.’ The hypothesis is reasonable. However, this hypothesis is not in accordance with the OT measurements. On one hand, the authors did not check whether the PB of Tomosyn would indeed affect the pairing of juxtamembrane regions between VAMP2 and Syx1; on the other hand, OT measurements confirmed that addition of Tomosyn attenuates the entry of SNAP25 into the template complex. The authors need to conduct more assays to solidify their conclusion, instead of inferring from physiological phenotypes.

8. The results that Tomosyn attenuates the entry of SNAP25 into the template complex seem to be inconsistent with previous observations that Tomosyn has no role in Munc18-Munc13-regulated SNARE complex assembly [Li et al., 2018, PMID: 29485200]. In other words, Tomosyn would attenuate Munc18-Munc13-guided SNARE complex assembly according to the authors’ results. Please present more discussions for this issue.

9. It would be helpful to conduct long-term recovery (i.e., 1 min or longer) for the data in Fig.3g, Fig. 6d and Fig. 7f as Fig. 2g did.

Minor comments:

1. Pg.11, line 34, should there be ‘Fig.7’ instead of ‘Fig.6’? Please check.
2. The methods utilized for analyzing synaptic Tomosyn level were not specified. Alternatively, non-standardized values are recommended if applying Mander’s correlation coefficient (MCC).

Reviewer #2 (Remarks to the Author):

This article examines the role of mammalian tomosyns1/2 in the regulation of SNARE complex assembly and neurotransmitter release. A comprehensive functional analysis of autapses in which vertebrate tomosyn1/2 was conditionally knocked-down (cDKO), suggests that tomosyn deficiency lowers the vesicle fusion barrier, thereby greatly increasing release probability and synaptic depression. These data

are consistent with fly data indicating that by modulating initial release probability, tomosyn allows for more sustained release within trains by limiting synaptic depression. Previous *in vitro* biochemical studies have attributed the actions of tomosyn to competition between the tomosyn SNARE domain and the synaptobrevin SNARE domain for fusogenic SNARE complex assembly. In contrast to this prevailing model, through a series of elegant single molecule force measurements using optical tweezers, the authors conclude that instead, tomosyn binds to synaptobrevin-containing template complexes preventing SNAP-25 binding during SNARE complex assembly. As such these findings provide important and provocative insights into the regulatory mechanism by which tomosyn acts on the release machinery. This article will be of great interest and significance to the scientific community and I highly recommend it for publication.

Comments:

I first want to commend the authors for the quality of the experimental data, the clarity and precision of the writing and the inclusion of really helpful model figures throughout.

The electrophysiological experiments were performed on cultured autaptic hippocampal neurons, an approach taken to circumvent complications encountered in constitutive mutant mouse models. Through a series of elegant and comprehensive experiments, Tomosyn cDKO neurons were shown to exhibit enhanced release (consistent with previous studies). Here tomosyn deficiency was attributed to a lowered vesicle fusion barrier leading to increased release probability. As noted by the authors, several aspects of the synaptic phenotype differ from those observed in invertebrate tomosyn mutants. Namely, the evoked release kinetics and sucrose responses are unaltered in the tomosyn cDKO autapses, as is SV docking. Although these may genuinely reflect differences between species, another possible reason for these differences could be due to the use of autaptic cultures in the present study to obtain the data, whereas the invertebrate data were obtained from NMJs, which arguably represent a more physiologically relevant environment. This concern has some basis from a previous study, which found significant differences in release properties when comparing hippocampal neuronal autapses to micro-islands of increasing synaptic connectivity (Liu et al 2009). Specifically, analysis of Syt1 KO mutants produced profoundly different findings: autaptic cultured neurons showing no changes in release, while cultures of multiple interconnected neurons (ranging from 2-10 neurons) showed significant reductions in both total evoked release and release probability. The latter is much closer to data from more intact preparations. Have the authors considered recording from small multicellular networks to determine whether the mammalian tomosyn cDKOs exhibit similar phenotypes to autapses.

A related question concerns the EM data that is presented in this study. Unlike *C. elegans* and fly NMJs, which exhibit a large increase in docked synaptic vesicles in tomosyn mutants, in cultured hippocampal neurons docking was completely unaltered. Another complication of this is that the EM analysis was not performed on autapses, but on micronetworks of 2-10 synapses. Comparing the docking within micronetworks to the functional analysis at autapses could potentially introduce flawed interpretations. Without knowing how tomosyn cDKOs affect small networks, comparing the EM from these to autaptic physiology raises some concern.

Minor points:

In extended Figure 1b are we seeing two replicates comparing control to cDKO? If so, are these replicates from the same or two different cultures? The figure legend states that the cDKO levels [of proteins] were normalized to controls....presumably after the control bands were normalized for loading with actin, correct? Could the authors comment on the finding that SNAP-25 levels are reduced by 20% in the tomosyn cDKOs compared to WT. How does this observation fit with the model presented in which SNAP-25 is less able to assemble into SNARE complexes in the wildtype?

Could the authors comment as to how tomosyns1/2 might reduce the Ca²⁺-sensitivity of synaptic responses.

In extended Figure 5, can the authors label the presynaptic density/active zone in the micrographs. Also in 5e (mislabelled as d in the figure legend), it is not clear how the vesicle distances were measured and binned into distinct increments. Do these represent concentric rings, or vectors from the center or edge of a morphologically defined active zone. Are both docked and cytoplasmic SVs binned together? It would be interesting to examine the distances of the docked SVs from the AZ separately, as this could reveal potential changes in docked SV distributions, as has been seen in other systems/mutants.

Reviewer #3 (Remarks to the Author):

This multidisciplinary study strongly supports a novel understanding of how tomosyn, a long-studied regulator of neurotransmitter release, functions at the molecular level. It has been proposed that tomosyn competes with the R-SNARE VAMP2 for binding to syntaxin/SNAP25, but recent advances have called the physiological relevance of syntaxin/SNAP25 complexes into question. Here, comprehensive analysis of synaptic transmission in engineered strains expressing wild-type or mutant tomosyns, complemented with careful single-molecule studies of tomosyn's role in SNARE assembly, lead to a new mechanistic model. Specifically, tomosyn binds to the so-called template complex – VAMP2 and syntaxin aligned on the surface of Munc18 – to block binding of the final Qb/c-SNARE SNAP25. The authors also find, quite unexpectedly, that the polybasic region at the C-terminus of tomosyn is functionally important, and they propose a plausible model. Together the results represent a major advance in our understanding of tomosyn function and, indeed, the central role of the template complex in synaptic vesicle fusion.

Overall, I can find little to criticize in this meticulously prepared manuscript. I think it will interest a broad audience of neuroscientists, cell biologists, and biophysicists and is therefore a strong candidate for publication in Nature Communications.

RESPONSE TO THE REFEREES' COMMENTS

We thank the referees for their positive and conscientious reports and unanimous support for publication. We are very pleased to see that the reviewers find the study of “great interest and significance to the scientific community” and appreciate the “new layer of understanding” and “important and provocative insights”. Below is a point-by-point reply to the specific concerns raised, the referees’ point in black and our response in blue.

REVIEWER COMMENTS

Reviewer #1

Major comments:

1. It is clear that the pair-pulse ratio (PPR) of Tomosyn-1/2 cDKO is lower than the control, which may denote that synapses depress faster in the absence of Tomosyns. However, the absolute EPSC amplitude and charge transfer of the second pulse (even the fourth pulse) of cDKO are larger than those of Control (Fig.1h-j). I am not firmly sure whether comparing the absolute values would make sense in such experiments. However, from a non-professional point of view, the synapse strength seems not decrease (i.e., the absolute amplitudes of cDKO are not ‘decreased’ compared to Control, Fig.1h, i), instead, it only decreases from a much higher level (i.e., hyperactive) to a ‘normal’ level. Please comment on the issues.

Response: We fully agree. The amount of depression and the difference in absolute responses during the first stimulus are a stronger phenotype than differences in absolute synaptic strength +/- tomosyns during subsequent pulses. Upon stronger stimulation (10Hz or higher), the absolute EPSC size in cDKO responses seems decreased compared to control (new fig 1h-I, Supplemental fig 2f). While addressing issues raised by reviewer 2 (see below), we also observe that after 5 stimuli, the absolute amplitude of the cDKO might be below the level of controls (Supplemental Fig. 3). However, we basically agree that in our interpretation, we have been a bit biased toward preserving synaptic vesicles. Important biological processes have positive and negative regulators and our results suggest that tomosyns are negative regulators, especially when release probability is highest (and that, consequently, depression rates are also altered). Stabilizing or equalizing synaptic strength during repetitive stimulation alters synaptic information processing and is probably a biologically relevant aspect of tomosyn phenotypes. To emphasize this, we have added this to the abstract (p. 1, l. 35-36).

2. In pg.3, lines 40-42, the authors wrote that ‘Thus, tomosyns lower the release probability of synapses, thereby reducing their initial strength and limiting synaptic depression.’ According to the early study on pair-pulse facilitation [Dobrunz and Stevens, 1997, PMID: 9208866], it is apparent that pair-pulse facilitation inversely correlates with release probability of pulse 1. The data presented in Fig.1f and g shows that the PPR of cDKO is lower than that of Control. The ‘pair-pulse facilitation’ in the above citation (also cited by the authors in the manuscript) is different to the ‘pair-pulse ratio (PPR)’ assessed in the manuscript, where the former is the ratio of release probability for the second pulse to the first pulse. Could these be equivalent? Please clarify.

Response: The reviewer is correct to point out that the calculations are different between the cited paper and the present manuscript. In Dobrunz and Stevens, release probability is defined as $1 - \text{the probability of no release}$ and facilitation is calculated as the ratio of the release probabilities of the two pulses. We chose to calculate the more intuitive and nowadays more widely used paired-pulse ratio by dividing the amplitudes of the two pulses (pulse2/pulse1) [e.g. in PMID: 11797009, PMID: 20400951, PMID: 27807164]. In this way, the value is above 1 when a cell facilitates and below 1 when it depresses. In addition, as pointed out by the reviewer, it allows to infer the degree of the

initial release probability because cells with a large initial release probability and a correspondingly large first amplitude generally depress while the opposite is true for cells with low initial release probability. Thus, the two methods of calculating release probability and facilitation differ but the same assumption of the inverse correlation applies. We edited the text to make this concept clearer for readers (p. 3, l. 38-41).

3. In Fig.1, the authors adopted 5 Hz stimuli, whereas 10 Hz stimuli was applied in Fig.3d, Fig. 6b, and Fig.7c. The authors termed them all as the 'STP'; meanwhile, the 'depression' degrees seem to be different when using different frequency. Please comment on the issue or present reasonable explanations.

Response: The reviewer is correct to note that we are presenting the STP measure calculated from a 5 Hz train in Fig. 1h-k but calculated from a 10 Hz train in all other figures mentioned. In our first experiment, which is shown in Fig. 1, we have applied multiple trains with different frequencies: 5 Hz, 10 Hz, and 20 Hz. We have calculated the STP measure for all of them and decided to present only the one calculated from the 5 Hz train in the main figure (and the rest in Supplementary Fig2 e/f). In the following experiments (shown in Fig. 3, 7 and 8) we decided to only apply the 10 Hz train. For better consistency, we now exchanged the data presented in Fig. 1 h-k with the ones calculated from the 10 Hz train (previously in Supplementary Fig. 2e) and placed the 5 Hz data in Supplementary Fig. 2e instead. Of course, the conclusions remain the same (p. 4, l. 2). We thank the reviewer for pointing out this inconsistency.

4. In pg.5, lines 28-29, 'Expressing tomosyn-1m... did not affect SV docking in cDKO cultures (+WT), but the number of docked SVs was reduced compared to control neurons.' It is unclear what the authors will express here. Please check the interpretation.

Response: We realize this sentence is not really clear. We have rephrased it. The interpretation (in the last sentence of the paragraph) remains the same (p. 6, l. 2-3).

5. The data in Fig.2g-i illustrated that Tomosyn is responsible for short-term (2 s) instead of long-term (1 min) synaptic recovery, according to the authors' interpretation. However, the absolute EPSC amplitudes between cDKO and Control in short-term recovery displayed no significant difference. On its face, does it represent that Tomosyn has no role in this short-term recovery pulse? Or can we assume that the short-term recovery (i.e., <2 s, or maybe longer) utilizes different mechanism that does not require the participation of Tomosyn and, if possible, Munc18s and/or other regulatory proteins? Certainly, the hypothesis is based on the premise that the augmented EPSC amplitude of the first pulse by Tomosyn-cDKO is abnormal (decrease the energy barrier of SV fusion as indicated by the authors). The authors seemed to present an arbitrary conclusion and did not conduct enough interpretations and discussions.

Response: We fully agree. We are well aware of this apparent divergence (and the fact that we do not really discuss it). Our favorite explanation has been that tomosyns do not regulate newly arriving vesicles and therefore the short-term recovery pulse has the same amplitude +/- tomosyns. In the end, we omitted this interpretation from the submitted manuscript because we felt it was too speculative. However, our new analyses (addressing point 9 below) with longer recovery pulses are consistent with this idea: after 90s, tomosyn-dependent suppression of P_{ves} is fully instated again. These new analysis and the reviewer's comment convinced us to include this interpretation in the revised manuscript (p. 13, l. 7-12).

6. It is interesting that Tomosyn would hinder the entry of SNAP25 into the template complex. Although the authors found that addition of Tomosyn induces a conformational change (1.5 nm greater than state 7) of the template complex through OT measurements, the interaction pattern between Tomosyn and the template complex is still quite unimaginable. One possible mechanism may

be that Tomosyn finitely competes the binding of VAMP2 to Munc18, since the C-terminal half of the SNARE motif of Tomosyn shares considerable homology with that of VAMP2. Meanwhile, the data in Fig. 6 (FA variant) also corroborate this. The authors should at least test the action of Tomosyn FA variant, Δ PB variants, Core-hybrid, LR-hybrid, and if possible, free VAMP2 C-half (or chimeric VAMP2-Tomosyn, or equivalents) in OT measurements to confirm the possible mechanism.

Response: Following reviewer's insightful suggestions, we have performed extensive experiments to investigate the molecular mechanism of the large conformational change observed by us. We made a series of tomosyn and SNARE modifications, including the ones suggested by the reviewer, and assessed their effects on tomosyn binding to the template complex. Our results indicate that a majority of the tomosyn SNARE motif contributes to its binding to the template complex. Overall, tomosyn and VAMP2 cooperate to form the tetrameric complex, as omitting either one failed to observe the complex. This cooperative association on the surface of Munc18-1 certainly involves extension interactions between tomosyn and Munc18-1. Although tomosyn and VAMP2 belong to the conserved R-SNARE family, they can be distinguished by intricate interactions between R-SNARE and Munc18-1. As a result, overall, they do not necessarily displace each other to bind Munc18-1. However, our results do not rule out that in certain small regions, the two proteins compete to bind Munc18-1. Although more experiments are required to elucidate the structure of the tomosyn-bound template complex, our preliminary data suggest that tomosyn binding induces a large conformational change in Munc18-1.

Based on these new data and understanding, we have added a new section titled "The tomosyn SNARE motif extensively binds to the template complex" with a new figure (Fig. 6 in the revised manuscript, p. 10, l. 1-27), which is not copied here.

7. It is clear that the poly-basic tail of Tomosyn play an important role in regulating neuronal SNARE complex assembly as shown in Fig.6. The authors concluded in the Discussion section that '..the SNARE motif and the polybasic domain of tomosyns might cooperatively interfere with C-terminal zippering of partially assembled SNARE complexes into the juxtamembrane domains of syb2/VAMP2 and syntaxin-1.' The hypothesis is reasonable. However, this hypothesis is not in accordance with the OT measurements. On one hand, the authors did not check whether the PB of Tomosyn would indeed affect the pairing of juxtamembrane regions between VAMP2 and Syx1; on the other hand, OT measurements confirmed that addition of Tomosyn attenuates the entry of SNAP25 into the template complex. The authors need to conduct more assays to solidify their conclusion, instead of inferring from physiological phenotypes.

Response: Following the reviewer's suggestion, we have conducted a new experiment to examine the impact of the poly-basic tail truncation on template complex formation and found that the truncation impairs the template complex (Fig. 6a, trace v). This observation corroborates the important role of the poly-basic tail in the regulated SNARE assembly, consistent with our *in vivo* results. Therefore, we have deleted the relevant paragraph to speculate its role in SNARE assembly.

8. The results that Tomosyn attenuates the entry of SNAP25 into the template complex seem to be inconsistent with previous observations that Tomosyn has no role in Munc18-Munc13-regulated SNARE complex assembly [Li et al., 2018, PMID: 29485200]. In other words, Tomosyn would attenuate Munc18-Munc13-guided SNARE complex assembly according to the authors' results. Please present more discussions for this issue.

Response: In Li's paper, the tomosyn's negative role in SNARE assembly was not emphasized but was evident in their data. For example, in Figs. 4B and 4D, the presence of tomosyn attenuated SNARE assembly in the presence of Munc18-1 and Munc13-1. Thus, our result is consistent with the result shown in Li's paper. However, the bulk assay used in this work cannot directly detect the template, nor the effect of tomosyn on the template complex.

To address the reviewer's comment, we have added the following sentence and citation in Discussion:

“Our finding is consistent with the negative role of tomosyn in SNARE assembly in the presence of both Munc18-1 and Munc13-1.” (p. 13, l. 29-30)

9. It would be helpful to conduct long-term recovery (i.e., 1 min or longer) for the data in Fig.3g, Fig. 6d and Fig. 7f as Fig. 2g did.

Response: To systematically test whether all mutant tomosyns restore the cDKO phenotype, we focused on two key aspects of the cDKO phenotype: enhanced depression and delayed recovery. The one minute recovery pulse was taken along in the original cDKO dataset to validate that cDKO neurons regain their initial EPSC size when given enough time. However, as both control and cDKO neurons had fully restored their EPSC size at 1 minute, this timepoint was not taken along for all tomosyn mutants. We agree that confirmation of intact ‘long-term’ recovery is interesting for the mutants. It would take a substantial amount of work (~9 months) and mice to do new experiments with all mutants (14 experimental groups). However, we realized that a similar recovery parameter can be extracted from the existing datasets. During the short 10Hz stimulus train, synapses in the cDKO depress to 50% of their EPSC size. A resting period of 90 seconds was applied before the next stimulus trains was given. We quantified the degree to which all 14 experimental groups recovered their EPSC to its initial size during this period and added the results in Supplemental Figure 7f, Supplemental Figure 10f and Supplemental Figure 11f. We found that all groups restored their EPSC size within this resting period, suggesting that neurons deficient in functional tomosyns regain their naïve (enhanced) synaptic strength within this time.

Minor comments:

1. Pg.11, line 34, should there be ‘Fig.7’ instead of ‘Fig.6’? Please check.

We thank the reviewer for pointing out this mistake. In fact, both Fig. 3 and Fig. 8 should be referenced here. We have corrected this in the text (p. 13, l. 41).

2. The methods utilized for analyzing synaptic Tomosyn level were not specified. Alternatively, non-standardized values are recommended if applying Mander’s correlation coefficient (MCC).

We apologize for the incomplete explanation and thank the reviewer for pointing this out. We have now added explanations both in the main text (p. 3, l. 19-24) and the methods section (p. 18, l. 6-8). In brief, average synaptic tomosyn levels were measured within the synapse mask created based on synaptophysin puncta using the analysis software SynD [PMID: 21167201]. Average intensity from tomosyn staining was measured within these puncta.

Reviewer #2:

Comments:

The electrophysiological experiments were performed on cultured autaptic hippocampal neurons, an approach taken to circumvent complications encountered in constitutive mutant mouse models. Through a series of elegant and comprehensive experiments, Tomosyn cDKO neurons were shown to exhibit enhanced release (consistent with previous studies). Here tomosyn deficiency was attributed to a lowered vesicle fusion barrier leading to increased release probability. As noted by the authors, several aspects of the synaptic phenotype differ from those observed in invertebrate tomosyn mutants. Namely, the evoked release kinetics and sucrose responses are unaltered in the tomosyn cDKO autapses, as is SV docking. Although these may genuinely reflect differences between species, another possible reason for these differences could be due to the use of autaptic cultures in the present study to obtain the data, whereas the invertebrate data were obtained from NMJs, which arguably represent a more physiologically relevant environment. This concern has some basis from a previous study, which found significant differences in release properties when comparing hippocampal neuronal autapses to micro-islands of increasing synaptic connectivity (Liu et al 2009). Specifically, analysis of Syt1 KO mutants produced profoundly different findings: autaptic cultured neurons showing no changes in release, while cultures of multiple interconnected neurons (ranging

from 2-10 neurons) showed significant reductions in both total evoked release and release probability. The latter is much closer to data from more intact preparations. Have the authors considered recording from small multicellular networks to determine whether the mammalian tomosyn cDKOs exhibit similar phenotypes to autapses.

Response: Of course, the invertebrate neuromuscular junction is a different model system from the autaptic neurons used in this study. It was not our intention to claim that the observed differences were specifically due to a species difference, and we adapted the discussion accordingly (p. 12, l. 21-23). We agree that recording from neurons in a network is an important confirmation of the conclusions reached in autapses. We tested tomosyn cDKO phenotypes in micro-networks of 3-10 neurons, as proposed by the reviewer. These experiments produced full confirmation of tomosyn cDKO phenotypes and in fact even stronger differences (Supplementary Fig. 3). Consistent with our findings in autapses, loss of tomosyns resulted in a strong increase in the frequency of spontaneous excitatory events in micro-networks (Autapse: WT 6.54 ± 6.54 Hz, cDKO 27.11 ± 13.87 Hz, Cohen's $d = -1.90$; Network: WT 5.35 ± 6.77 Hz, cDKO 37.39 ± 19.02 Hz, Cohen's $d = -2.24$). We then recorded excitatory currents evoked by local stimulation via a bipolar electrode. Tomosyn deficiency resulted in a larger 1st evoked and stronger synaptic depression during short stimulus trains, fully in line with our results in autapses. Hence, tomosyns reduce the release probability equally in autaptic and small network cultures. (p. 4, l. 4-11)

Since evoking excitatory currents in networks can be complicated by occasional network activation, most studies focus on the characterization of inhibitory currents (Maximov et al., 2007, PMID: 17118459); Kaeser et al., 2011 (PMID: 21241895); Courtney et al., 2019 (PMID: 31501440)). When optimizing our protocol, we observed that a) short-term plasticity is highly reproducible within a micro-network and b) a stimulus intensity of 1.5nA produced largely monosynaptic excitatory currents in the presence of picrotoxin. Although EPSCs were occasionally contaminated with secondary peaks from network activation, this did not affect the measurement of the fast inward currents typical for AMPA-type EPSCs (see also Maximov et al., 2009; PMID: 19164751). We tried sustained high-freq stimulation to deplete the vesicle pool but this resulted in electrolysis of the electrode tip which interfered with the stimulus.

A related question concerns the EM data that is presented in this study. Unlike *C. elegans* and fly NMJs, which exhibit a large increase in docked synaptic vesicles in tomosyn mutants, in cultured hippocampal neurons docking was completely unaltered. Another complication of this is that the EM analysis was not performed on autapses, but on micronetworks of 2-10 synapses. Comparing the docking within micronetworks to the functional analysis at autapses could potentially introduce flawed interpretations. Without knowing how tomosyn cDKOs affect small networks, comparing the EM from these to autaptic physiology raises some concern.

Response: We performed the requested experiment in micro-networks and confirmed that loss of tomosyns affects small networks in the same way as autapses (see above). Hence, our main conclusion that the cDKO phenotype is unlikely to be explained by an increased docked vesicle pool remains unaltered. We have now included a second EM dataset, originally generated for a different question (on the other secretory pathway, the release of neuromodulators from dense core vesicles, PMID: 37695731) to test docking phenotypes for larger neuronal networks. This analysis confirmed that the docked vesicle pool and vesicle distribution were not affected, indicating that the use of different neuronal cultures does not influence the cDKO phenotype. (p. 6, l. 7-11)

Minor points:

In extended Figure 1b are we seeing two replicates comparing control to cDKO? If so, are these replicates from the same or two different cultures? The figure legend states that the cDKO levels [of proteins] were normalized to controls....presumably after the control bands were normalized for loading with actin, correct?

Response: We are sorry for the missing explanation and thank the reviewer for pointing this out. The two replicates that are on the Western blot picture in Supplementary Fig. 1b are two technical replicates from one culture. However, the quantification shown in 1c was done on four independent cultures (i.e. biological replicates). Each culture was run in three technical replicates and the mean of the three technical replicates for every culture is shown as a circle on the graph. The bars show the means over all biological replicates. The bands for STXBP5 were first normalized to the actin band before normalizing to controls (i.e. calculating the percentage of control). We have added this information to the methods section (p. 17, l. 28-31) and the figure legend.

Could the authors comment on the finding that SNAP-25 levels are reduced by 20% in the tomosyn cDKOs compared to WT. How does this observation fit with the model presented in which SNAP-25 is less able to assemble into SNARE complexes in the wildtype?

Response: This is indeed an interesting observation. It is tempting to speculate that because SNAP25 cannot enter the SNARE-complex it becomes more susceptible to degradation as observed for other synaptic proteins that cannot form their native multimeric complexes. However, this would result in higher, not lower, SNAP-25 levels in the tomosyn cDKO. Alternatively, the higher basal rate of SNARE complex assembly/disassembly due to the increase mini frequency in cDKO neurons might render SNAP-25 vulnerable to misfolding and subsequent proteasomal degradation (Sharma et al., 2011, PMID: 21151134). However, we felt the effect was too small and the idea too speculative to include it in the discussion, but if the reviewer and editor think it is important enough, we are happy to include it.

Could the authors comment as to how tomosyns1/2 might reduce the Ca²⁺-sensitivity of synaptic responses.

In the presence of tomosyns, a stronger trigger such as a higher calcium concentration is needed to release the same amount of vesicles. In a dose-response curve in which EPSC size is plotted against extracellular [Ca²⁺], this manifests as a shift in the apparent calcium sensitivity of synaptic responses. However, tomosyns also reduced the amount of vesicles released triggered by a calcium-independent osmotic shock. Based on both observations, we concluded that tomosyns reduce the probability that vesicles will fuse without changing the properties of the calcium sensor(s) for fusion.

In extended Figure 5, can the authors label the presynaptic density/active zone in the micrographs. Also in 5e (mis-labeled as d in the figure legend), it is not clear how the vesicle distances were measured and binned into distinct increments. Do these represent concentric rings, or vectors from the center or edge of a morphologically defined active zone. Are both docked and cytoplasmic SVs binned together? It would be interesting to examine the distances of the docked SVs from the AZ separately, as this could reveal potential changes in docked SV distributions, as has been seen in other systems/mutants.

The distribution of active zone proximal vesicles (both docked and undocked vesicles) was measured in synapse profiles as the shortest distance from the vesicular membrane to the plasma membrane within active zone region and plotted in 5-10 nm bins. We have now added this explanation to the material and methods (p. 20, l. 1-3). Synaptic vesicle docking is defined as no observable distance (less than 0.7 nm) between the vesicular membrane and the active zone, thus no distance distribution profile to the active zone can be produced of docked vesicles alone. We have additionally analyzed vesicle contacting the plasma membrane outside the active zone region, analogous to analysis in invertebrates [Gracheva et al 2006 PMID: 16895441, Sauvola et al 2021 PMID: 34713802], which is incidentally detected in mouse synapse cross sections (control: 26 of 130 synapses, cDKO: 11 of 88 synapses, cDKO+Tom: 12 of 126 synapses). The average number of contacting vesicles outside active zone is not different between groups (average \pm SEM: control: 0.415 \pm 0.091, cDKO: 0.206 \pm 0.069, cDKO+Tom: 0.159 \pm 0.051; multi-level $p = 0.21$; now included in table 2 and main text), but too few

vesicles were found to make a meaningful distribution. Given the lack of difference between groups and the small pool size, we do not think these vesicles can explain the major EPSC amplitude effects observed in tomosyn cDKO neurons.

Reviewer #3:

This multidisciplinary study strongly supports a novel understanding of how tomosyn, a long-studied regulator of neurotransmitter release, functions at the molecular level. It has been proposed that tomosyn competes with the R-SNARE VAMP2 for binding to syntaxin/SNAP25, but recent advances have called the physiological relevance of syntaxin/SNAP25 complexes into question. Here, comprehensive analysis of synaptic transmission in engineered strains expressing wild-type or mutant tomosyns, complemented with careful single-molecule studies of tomosyn's role in SNARE assembly, lead to a new mechanistic model. Specifically, tomosyn binds to the so-called template complex – VAMP2 and syntaxin aligned on the surface of Munc18 – to block binding of the final Qb/c-SNARE SNAP25. The authors also find, quite unexpectedly, that the polybasic region at the C-terminus of tomosyn is functionally important, and they propose a plausible model. Together the results represent a major advance in our understanding of tomosyn function and, indeed, the central role of the template complex in synaptic vesicle fusion.

Overall, I can find little to criticize in this meticulously prepared manuscript. I think it will interest a broad audience of neuroscientists, cell biologists, and biophysicists and is therefore a strong candidate for publication in Nature Communications.

Response: We thank the reviewer for his/her kind words, the attention given to our manuscript and strong support.

General point

During the resubmission phase for this manuscript, a manuscript describing another study from our lab was accepted for publication (<https://elifesciences.org/articles/85561>) that addresses the role of tomosyn in the other main secretory pathway, for neuromodulators released from dense core vesicles (DCVs), using tomosyn double null mutant neurons. We feel it is relevant to discuss these data in the current manuscript too, for comparison between the two main secretory pathways and also because the role of tomosyns in DCV exocytosis is in fact quite limited and different from the role we describe here for tomosyns in synaptic vesicle exocytosis. These notions have now been added to the Discussion in the revised manuscript (p. 12, l. 17-19).

REVIEWERS' COMMENTS

Reviewer #1 (Remarks to the Author):

The authors have directly addressed the concerns raised before. The reviewer only has some minor comments on the manuscript:

1. For the data in Fig.6, especially Fig.6a(v), the authors concluded that '.. even the C-terminal polybasic domain of tomosyn enhanced binding, as its truncation reduced the lifetime and probability of tomosyn binding (trace v).' It is not easy (for the readers) to extract the information of 'reduced lifetime' from the trace shown in the right panel. The authors may provide a clearer visualization of the data, e.g., to make histograms for the critical parameters, and/or list a table to accommodate the critical parameters extracted from the traces.
2. Meanwhile, the OT data with only example traces and force-extension curve (FEC) are overspecialized for non-experts. The authors may need to make more efforts to present reader-friendly visualizations as suggested in point#1.

Reviewer #2 (Remarks to the Author):

I appreciate the efforts that have been made to address the reviewers comments and believe these have addressed any concerns that I had regarding the data. I am strongly in favor of publishing this important and rigorous study.

as a minor comment, although I think the inclusion of EM docking data from chemically fixed culture in supplementary figure 6f-h is not the preferred method for analyzing vesicle docking, the HPF data does appear to show no change in the number of docked SVs in the tomosyn dKOs and is therefore sufficient.

RESPONSE TO THE REFEREES' COMMENTS

We thank the referees for their contributions to our manuscript and are very pleased that we managed to address all their previous concerns. Below is a point-by-point reply to the minor comments that remained, the referees' point in black and our response in blue.

REVIEWERS' COMMENTS

Reviewer #1

The authors have directly addressed the concerns raised before. The reviewer only has some minor comments on the manuscript:

1. For the data in Fig.6, especially Fig.6a(v), the authors concluded that '.. even the C-terminal polybasic domain of tomosyn enhanced binding, as its truncation reduced the lifetime and probability of tomosyn binding (trace v).' It is not easy (for the readers) to extract the information of 'reduced lifetime' from the trace shown in the right panel. The authors may provide a clearer visualization of the data, e.g., to make histograms for the critical parameters, and/or list a table to accommodate the critical parameters extracted from the traces.

Response: Following the reviewer's suggestion, we have calculated the histogram distributions of dwell time in the tomosyn-bound template complex state and showed a close-up view of the three traces containing the tomosyn-bound template complex state (Fig. 6a, traces i, ii, and v), resulting in the new Supplementary Figure 10. The results of our calculations support our conclusion that tomosyn truncation reduces the average lifetime of the tomosyn-bound state.

2. Meanwhile, the OT data with only example traces and force-extension curve (FEC) are overspecialized for non-experts. The authors may need to make more efforts to present reader-friendly visualizations as suggested in point#1.

Response: We agree with the reviewer that some of our trajectories are too crowded. Following the reviewer's suggestion, we have provided a close-up view of key trajectories in the new Supplementary Figure 10. We have also added or revised a few sentences and references to clarify force-extension curves and other aspects of data, including

"This FEC is equivalent to a phase diagram revealing the states and their transitions of the tomosyn SNARE complex as a function of force and extension."

"Interestingly, a distinct and irreversible extension decrease appeared, which only started from the extension corresponding to the template complex, but not the tomosyn-bound template complex (N=28, Fig. 5b, FEC#3; Fig. 5d, bottom trace)."

We hope these changes help clarify the descriptions of our results.

Reviewer #2

I appreciate the efforts that have been made to address the reviewers comments and believe these have addressed any concerns that I had regarding the data. I am strongly in favor of publishing this important and rigorous study.

as a minor comment, although I think the inclusion of EM docking data from chemically fixed culture in supplementary figure 6f-h is not the preferred method for analyzing vesicle docking, the HPF data does appear to show no change in the number of docked SVs in the tomosyn dKO and is therefore sufficient.

Response: We agree that HFP is the preferred method to analyze vesicle docking and will follow the reviewers suggestion to not include the chemically fixed EM data. Hence, we have removed panel f-h from Supplementary Figure 6.